# Understanding Reconstruction Attacks with the Neural Tangent Kernel and Dataset Distillation

**Noel Loo, Ramin Hasani, Mathias Lechner, Alexander Amini and Daniela Rus**
MIT CSAIL
Cambridge, Massachussetts, USA
`{loo, rhasani, mlechner, amini, rus}@mit.edu`

## Abstract

Modern deep learning requires large volumes of data, which could contain sensitive or private information that cannot be leaked. Recent work has shown for homogeneous neural networks a large portion of this training data could be reconstructed with only access to the trained network parameters. While the attack was shown to work empirically, there exists little formal understanding of its effective regime and which datapoints are susceptible to reconstruction. In this work, we first build a stronger version of the dataset reconstruction attack and show how it can provably recover the *entire training set* in the infinite width regime. We then empirically study the characteristics of this attack on two-layer networks and reveal that its success heavily depends on deviations from the frozen infinite-width Neural Tangent Kernel limit. Next, we study the nature of easily-reconstructed images. We show that both theoretically and empirically, reconstructed images tend to "outliers" in the dataset, and that these reconstruction attacks can be used for *dataset distillation*, that is, we can retrain on reconstructed images and obtain high predictive accuracy.[1]

## 1 Introduction

Neural networks have been shown to perform well and even generalize on a range of tasks, despite achieving zero loss on training data (Zhang et al., 2017; 2021). But this performance often cannot be realized in practice due to security constraints in deployment. A fundamental question in the security of neural networks is how much information is *leaked* via this training procedure, that is, can adversaries with access to trained models, or predictions from a model, infer what data was used to train the model? Ideally, we want to ensure that our models are resistant to such attacks. However, in practice, we see that this ideal is commonly violated. One heinous violation of this principle is the phenomenon of memorization (Arpit et al., 2017; Feldman & Zhang, 2020b; Feldman, 2020; Carlini et al., 2019), where trained networks can be shown to replicate their training data at test time in generative models. A more extreme example of memorization is presented in Haim et al. (2022), where the authors show that it is possible to recover a large subset of the training data given only the trained network parameters.

The existence of this attack begs many follow-up questions: "*Under what circumstances is this attack successful?*"; and "*What are properties of these recovered images?*" In this paper, we consider a stronger variant of the attack presented in Haim et al. (2022), provide novel theoretical and empirical insights about dataset reconstruction attacks, and provide answers to the above questions. In particular, we make the following new **contributions**:

**We design a stronger version of Haim et al. (2022)'s dataset reconstruction attack** that can provably reconstruct the *entire* training set for networks in the neural tangent kernel (NTK) (Jacot et al., 2018) regime when trained under mean squared error (MSE) loss. This attack transfers to finite networks with its success dependent on deviations from the NTK regime.

---

[1]Code is available at `https://github.com/yolky/understanding_reconstruction`

**We show that outlier datapoints are prone to reconstruction** under our attack, corroborating prior work observing this property. Additionally, we show that removing easily reconstructed images can improve predictive accuracy.

**We formally prove and empirically show that a dataset reconstruction attack is a variant of *dataset distillation*.** The reconstruction loss is equal to the loss of the kernel-inducing points (KIP) (Nguyen et al., 2021a;b) dataset distillation algorithm, under a different norm, plus a variance-controlled term. Furthermore, we can retrain models using recovered images and achieve high performance.

## 2 BACKGROUND AND RELATED WORKS

**Machine Learning Privacy.** A large body of work studies how to extract sensitive information from trained models. This is problematic as legislation such as HIPAA and GDPR enforce what data can and cannot be published or used (Centers for Medicare & Medicaid Services, 1996; European Commission, 2016). Quantifying the influence of training examples leads to the topic of influence functions (Koh & Liang, 2017), and Membership-inference attacks (Shokri et al., 2016; Carlini et al., 2021), which try to infer whether particular examples were used in training. Defending against these attacks is the study of *differential privacy*, which quantifies and limits the sensitivity of models to small changes in training data (Dwork et al., 2006; Abadi et al., 2016). Likewise, the field of *machine unlearning* tries to remove the influence of training examples post-training (Bourtoule et al., 2019). Without defense techniques, trained networks have been shown to leak information (Rigaki & Garcia, 2020). For generative language models, (Carlini et al., 2019) show large language models often reproduce examples in the training corpus verbatim. *Model inversion* techniques aim to recreate training examples by looking at model activations (Fredrikson et al., 2015; Yang et al., 2019; He et al., 2019). This memorization phenomenon can be shown to be necessary to achieve high performance under certain circumstances (Feldman & Zhang, 2020b; Brown et al., 2021).

**Dataset Reconstruction.** Recent work (Haim et al., 2022) has shown that one can reconstruct a large subset of the training data from trained networks by exploiting the implicit biases of neural nets. They note that homogeneous neural networks trained under a logistic loss converge in direction to the solution of the following max-margin problem (Lyu & Li, 2020; Ji & Telgarsky, 2020):

$$\arg\min_{\theta'} \frac{1}{2}\|\theta'\|_2^2 \quad \text{s.t.} \quad \forall i \in [n], y_i f_{\theta'}(x_i) \geq 1, \tag{1}$$

Where $\{x_i, y_i\}$ is the training set with images $x_i$ and labels $y_i \in \{+1, -1\}$, and $f_{\theta'}(x)$ the neural network output with parameters $\theta'$. (Haim et al., 2022) shows that by taking a trained neural network and optimizing images (and dual parameters) to match the Karush–Kuhn–Tucker (KKT) conditions of the max-margin problem, it is possible to reconstruct training data. This is an attack that causes leakage of training data. Here, we consider a stronger variant of the attack that requires training under mean-squared error (MSE) loss.

**Neural Tangent Kernel.** To investigate the generalization in neural networks we can use the neural tangent kernel (NTK) theory (Jacot et al., 2018; Arora et al., 2019). NTK theory states that networks behave like first-order Taylor expansions of network parameters about their initialization as network width approaches infinity (Lee et al., 2019). Furthermore, the resulting feature map and kernel converge to the NTK, and this kernel is frozen throughout training (Jacot et al., 2018; Arora et al., 2019). As a result, wide neural networks are analogous to kernel machines, and when trained with MSE loss using a support set $X_S$ with labels $y_S$ result in test predictions given by:

$$\hat{y}_T = K_{TS}K_{SS}^{-1}y_S,$$

with $K$ being the NTK. For fully-connected networks, this kernel can be computed exactly very quickly (as they reduce to arc-cosine kernels), but for larger convolutional networks, exact computation slows down dramatically (Arora et al., 2019; Zandieh et al., 2021). In practice, it has been shown that networks often deviate far from the frozen-kernel theoretical regime, with the resulting empirical NTKs varying greatly within the first few epochs of training before freezing for the rest (Hanin & Nica, 2020; Aitken & Gur-Ari, 2020; Fort et al., 2020; Loo et al., 2022b; Tsilivis & Kempe, 2022). In this paper, we use the NTK theory to gain a better understanding of these reconstruction attacks.

**Dataset Distillation.** Dataset distillation aims to construct smaller synthetic datasets which accurately represent larger datasets. Specifically, training on substantially smaller *distilled dataset* achieves performance comparable to the full dataset, and far above random sampling of the dataset (Wang et al., 2018; Zhao et al., 2021; Zhao & Bilen, 2021; Nguyen et al., 2021a;b; Zhou et al., 2022; Loo et al., 2022a). There are many algorithms for this, ranging from methods that directly unroll computation (Wang et al., 2018), try to efficiently approximate the inner unrolled computation associated with training on distilled data (Zhou et al., 2022; Loo et al., 2022a; Nguyen et al., 2021b), and other heuristics (Zhao et al., 2021; Zhao & Bilen, 2021). One algorithm is kernel-induced points (KIP) (Nguyen et al., 2021a;b), which leverages NTK theory to derive the following loss:

$$\mathcal{L}_{KIP} = \frac{1}{2}\|y_t - K_{TS}K_{SS}^{-1}y_S\|_2^2.$$

The loss indicates the prediction error of infinite width networks on distilled images $X_S$ and labels $y_S$, which are then optimized. We bring up dataset distillation as we show in this paper that our dataset reconstruction attack is a generalization of KIP, and that dataset distillation can be used to defend against the attack.

## 3 A NEURAL TANGENT KERNEL RECONSTRUCTION ATTACK

Haim et al. (2022) considers the scenario where the attacker only has access to the final trained network parameters. This attack requires that the networks are homogeneous and are trained for many epochs until convergence so that the network converges in direction to the final KKT point of Eq. 1. While it is a good proof-of-concept for such attacks, there are several theoretical and practical limitations of this attack. We find that the attack presented in Haim et al. (2022) is brittle. Namely, we were unable to reliably reproduce their results without careful hyperparameter tuning, and **careful network initialization strategies**. Their attack also requires training until directional convergence, which **requires network parameters to tend to infinity**, and requires homogenous networks. Furthermore, outside of the top few reconstructions, the overwhelming majority of reconstructions ($> 70\%$) are of poor quality (more in-depth discussions in appendix A). Here we present an attack which is compatible with early stopping, does not require special initialization strategies, and can provable reconstruct the *entire training set* under certain assumptions, the **first guarantee of reconstruction in any regime**. However, our attack requires access to the model initialization or a previous training checkpoint. Access to the initialization or earlier training checkpoints arises naturally in many settings, such as fine tuning from public models, or in federated learning where clients receive period updates of the model parameters.

Note that we cannot compare our attack to other ones such as gradient leakage attacks (Zhu et al., 2019), membership inference attacks (Shokri et al., 2016) and generative model attacks (Carlini et al., 2020), as these attacks either require gradient access (which also requires parameter access) in the setting on gradient leakage, specific query data points in the case of membership inference, or a generative model and query points for generative model attacks. Our attack requires no a priori knowledge of the dataset. With this context in mind, the attack desribed in this paper contributes to the literature on parameter-only based attacks.

Consider a neural network trained under MSE loss, $\mathcal{L} = \frac{1}{2}\sum_{i=0}^{N-1}(y_i - f_\theta(x_i))^2$, for $x_i, y_i \in X_T, y_T$, being the training set datapoints and labels. Now further assume that the network is trained under gradient flow and that the network is approximately in the lazy/NTK regime, that is, it behaves like a first-order Taylor expansion of the network outputs (Chizat et al., 2019):

$$f_\theta(x) \approx f_{lin,\theta}(x) = f_{\theta_0}(x) + (\theta - \theta_0)^\intercal \nabla_\theta f_{\theta_0}(x) \tag{2}$$

Lee et al. (2019) shows that the time evolution of the network parameters in this regime is given by:

$$\theta(t) = \theta_0 - \nabla_\theta f_{\theta_0}(X_T)^\intercal K_0^{-1}\left(I - e^{-\eta K_0 t}\right)(f_{\theta_0}(X_T) - y_T)$$

With $\eta$ the learning rate and $K_0$ the finite-width/empirical NTK evaluated at $\theta_0$. Namely, the final change in parameters is given by:

$$\Delta\theta = \theta_f - \theta_0 = \nabla_\theta f_{\theta_0}(X_T)^\intercal K_0^{-1}(y_T - f_{\theta_0}(X_T)) \tag{3}$$

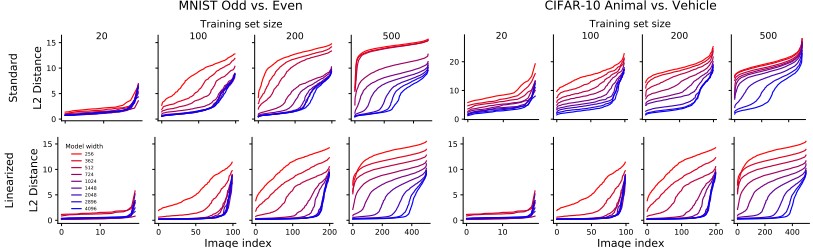

Figure 1: Reconstructed images (top) vs closest training images (bottom) for MNIST Odd vs. Even, and CIFAR-10 Animal vs. Vehicle Classification. Reconstructions were made from 4096-width two hidden layer fully-connected networks trained with standard dynamics and low learning rates. Apart from small amounts of noise, the original images and their reconstructions are visually indistinguishable.

Figure 2: Reconstruction quality curves for the MNIST Odd vs. Even and for CIFAR-10 Animal vs. Vehicle classification. We either used standard dynamics (top) or linearized dynamics (bottom) and varied both the size of the training set and model width. Smaller datasets are easier to reconstruct while wider models can reconstruct more images, with linearization helping in both scenarios.

Notably, this is the solution to the following optimization problem:

$$\underset{\Delta\theta}{\arg\min} \frac{1}{2}\|\Delta\theta\|_2^2 \quad \text{s.t.} \quad \Delta\theta^\mathsf{T}\nabla_\theta f_{\theta_0}(X_T) = y_T - f_{\theta_0}(X_T) \tag{4}$$

The corresponding KKT conditions are:

$$\Delta\theta = \alpha^\mathsf{T}\nabla_\theta f_{\theta_0}(X_T) \tag{5}$$
$$\Delta\theta^\mathsf{T}\nabla_\theta f_{\theta_0}(X_T) = y_T - f_{\theta_0}(X_T) \tag{6}$$

With $\alpha$ being the set of dual parameters. In our formulation, eq. (5) ensures we are at a stationary point, while eq. (6) ensures that the labels are correct. Like with (Haim et al., 2022), we can directly optimize the reconstruction images and dual parameters ($X$ and $\alpha$, respectively) to match these KKT conditions, given a network's final parameters and initialization to get $\Delta\theta$. In practice, we only need to optimize eq. (5), for reasons we will describe next, leading to our reconstruction loss:

$$\mathcal{L}_{\text{Reconstruction}} = \|\Delta\theta - \alpha^\mathsf{T}\nabla_\theta f_{\theta_0}(X_T)\|_2^2 \tag{7}$$

**Reconstruction in Infinite Width.** Next, we show that this formulation of the attack recovers the *entire* training set for infinite-width models. We further assume that the training data lies on the unit hypersphere.

**Theorem 1.** *(Informal) If $\mathcal{L}_{reconstruction} = 0$ (from Eq. 7), then we reconstruct the entire training set in the infinite-width limit, assuming that training data lies on the unit hypersphere.*

The proof works by considering the limiting case of the reconstruction loss in infinite width. By taking limits, we can convert the reconstruction loss into the Maximum Mean Discrepancy (MMD) (Gretton et al., 2012) comparing two measures: one parameterized by our training images, and the other by our reconstruction images, where the MMD is taken w.r.t the Neural Tangent Kernel (NTK). Using results from (Jacot et al., 2018; Lee et al., 2019), we know that these kernels converge in probability to the fixed infinite-width NTK, which is universal over datapoints on the unit hypersphere. This universality implies that $\mathcal{L}_{\text{reconstruction}} = \text{MMD} = 0$ iff $X_R = X_T$, meaning we recover the original training set. We refer the reader to appendix B for details and formal statement of theorem 1.

Note that in practice we do not enforce the unit sphere requirement on the data, and we still see high reconstruction quality, which we show in section 4. This mapping from the network tangent space to

Table 1: Distillation test accuracy of KIP, Recon-KIP (RKIP), RKIP from a trained network (RKIP-finite), on distilling 500 images down to 20 images. KIP and RKIP provide the best infinite-width performance, while KIP fails for finite models. (n=7)

| Distillation Algorithm | MNIST Odd/Even | | | CIFAR-10 Animal/Vehicle | | |
|---|---|---|---|---|---|---|
| | Standard | Linearized | Infinite Width | Standard | Linearized | Infinite Width |
| Full dataset (500 images) | $92.85 \pm 0.42$ | $92.91 \pm 0.33$ | $93.18 \pm 0.37$ | $75.06 \pm 0.21$ | $74.60 \pm 0.21$ | $75.42 \pm 0.28$ |
| KIP | $57.42 \pm 8.41$ | $55.62 \pm 7.48$ | $\mathbf{91.53 \pm 0.57}$ | $35.26 \pm 5.67$ | $32.37 \pm 3.60$ | $70.98 \pm 0.43$ |
| RKIP | $\mathbf{89.61 \pm 1.18}$ | $\mathbf{89.99 \pm 1.11}$ | $91.44 \pm 0.48$ | $\mathbf{72.23 \pm 3.61}$ | $\mathbf{72.76 \pm 3.74}$ | $\mathbf{74.66 \pm 0.93}$ |
| RKIP-finite | $88.45 \pm 0.89$ | $86.15 \pm 3.39$ | $87.31 \pm 3.24$ | $71.96 \pm 1.14$ | $63.99 \pm 4.02$ | $62.05 \pm 4.17$ |
| Random images | $73.52 \pm 3.60$ | $73.54 \pm 3.61$ | $74.12 \pm 3.73$ | $70.36 \pm 2.53$ | $70.18 \pm 2.54$ | $70.77 \pm 2.04$ |

image space also sheds light on the success of gradient leakage attacks (Zhu et al., 2019), in which gradients are used to find training batch examples.

## 4 DATASET RECONSTRUCTION FOR FINITE NETWORKS

While the attack outlined in Theorem 1 carries fundamental theoretical insights in the infinite-width limit, it has limited practicality as it requires access to the training images themselves to compute the kernel inner products. How does the attack work for finite-width neural networks, and under what circumstances is this attack successful?

To answer these questions, we follow the experimental protocol of Haim et al. (2022), where we try to recover images from the MNIST and CIFAR-10 datasets on the task of odd/even digit or animal/vehicle classification for MNIST and CIFAR-10, respectively. We vary the size of the training set from 10 images per class to 250 images per class (500 total training set size). We consider two hidden layer neural networks with biases using standard initialization (as opposed to NTK parameterization or the initialization scheme proposed in

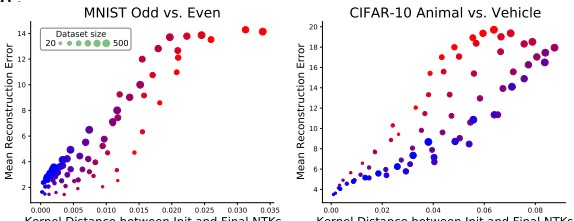

Figure 3: Mean reconstruction error vs. the kernel distance from the initialization to the final kernel. The mean reconstruction error, measured as the average value of the reconstruction curve, is strongly correlated with how much the finite-width NTK evolves over training. Dataset size is given by dot size, while the color indicates model width (see fig. 2).

Haim et al. (2022)). We vary the width of the neural networks between 256 and 4096 to see how deviations from the infinite-width regime affect the reconstruction quality. Furthermore, it is known that for finite-width networks the finite-width NTK varies over the course of training, deviating from the infinite-width regime. We can force the kernel to be frozen by considering *linearized* training, where we train a first-order Taylor expansion of the network parameters around its initialization (see eq. (2)). We consider both networks under standard (unmodified) dynamics and linearized dynamics. In appendix L.1 and appendix L.2 we consider convolutional architectures and high-resolution datasets, respectively, but we restrict our attention to MLPs on lower-resolution images in the main text.

We train these networks for $10^6$ iteration using full-batch gradient descent with a low learning rate, and during the reconstruction, we make $M = 2N$ reconstructions with $N$ being the training set size. A full description of our experimental parameters is available in appendix J and algorithmic runtime details in appendix D. To measure reconstruction quality we consider the following metric. We first measure the squared $L_2$ distance in pixel space from each training image to each reconstruction. We select the pair of training images and reconstruction which has the lowest distance and remove it from the pool, considering it pair of image/reconstruction. We repeat this process until we have a full set of $N$ training images and reconstructions (See Fig. 1). We then order the $L_2$ distances into an ascending list of distances and plot this function. We call this the *reconstruction curve* associated with a particular reconstruction set. We plot these reconstruction curves for varying dataset sizes and model widths in fig. 2. From fig. 2 we have the following three observations:

**Smaller training sets are easier to reconstruct.** We see that the reconstruction curve for smaller datasets has low values for all model widths. **Wider models can resolve larger datasets.** We

Figure 5: Reconstruction curves for networks trained on multiclass MNIST/CIFAR-10.

observe that for a given model width, there is a threshold image index at which the quality of reconstructions severely decreases. For example, for MNIST Odd/Even, 200 images and a width of 1024, this is 80 images. As we increase the model width this threshold increases almost monotonically.

**Linearization improves reconstruction quality.** We see that linearized networks can resolve more images and have better images compared to their same-width counterparts. The success of linearization suggests that deviations from the frozen NTK regime affect reconstruction quality. We can measure the deviation from the frozen kernel regime by measuring the *kernel distance* of the network's initialization NTK and its final NTK, given by the following: $d(K_0, K_f) = 1 - \frac{\text{Tr}(K_0^\intercal K_f)}{\|K_0\|_F \|K_f\|_F}$.

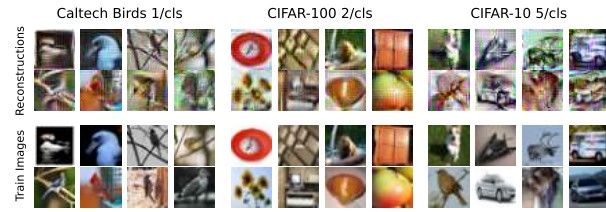

Figure 4: Reconstructions of training data for few-shot fine tuning on a ResNet-18 pretrained on ImageNet on Caltech Birds (1/cls), CIFAR-100 (2/cls) and CIFAR-10 (5/cls).

Intuitively, this distance tells us how well the initialization and final kernel align. Large values indicate that the kernel has changed substantially, meaning the network is deviating far from the NTK regime. We plot these kernel distances against the mean value of the reconstruction curves in figure fig. 3. We see immediately that reconstruction quality is strongly correlated with kernel distance, and that smaller datasets and wider models have a lower kernel distance. In appendix I, we discuss how our attack is **compatible with early stopping and cross-entropy loss**, unlike (Haim et al., 2022), which requires training until convergence. A more detailed discussion of the effect of early stopping is available in appendix I.

**Multiclass Classification.** In previous sections, we showed the validity of the attack on binary classification. Here we verify that the attack works with multiple classes. We repeat the same procedure as in section 4, but will all 10 classes. Details are given in appendix D, as well as additional results on 200-way classification on Tiny-ImageNet in appendix L.2. Results are shown by reconstruction curves in fig. 5. We observe that this attack has **improved** reconstruction quality with more classes. In appendix L.3, we observe that multi-class classification leads to lower kernel distances, suggesting it behaves more in the kernel regimes, explaining the better reconstruction quality. Future work could investigate this further.

## 5 Dataset Reconstruction in Fine Tuning

A key requirement of the attack is the model initialization. When training from scratch, attackers will not have access to this, making the attack useless. However, practitioners often do not train from scratch, but rather fine-tune large publicly available pre-trained models. Furthermore, users often do not have access to large amounts of data, effectively making the task few-shot. With evidence suggesting that training neural networks later during fine-tuning is well approximated by the frozen finite-NTK theory (Zancato et al., 2020; Shon et al., 2022; Zhou et al., 2021; Malladi et al., 2023), this makes the few-shot fine-tuning setting an easy target for this attack. To evaluate our attack in this setting, we fine-tuned publically available ResNet-18s pretrained on ImageNet on few-shot image classification on Caltech Birds (200 classes) (Welinder et al., 2010), CIFAR-100, CIFAR-10 with 1, 2, and 5 samples per class, respectively. For $\theta_0$, we use the initial fine-tuned model parameters. We see the best reconstructions in fig. 4. We see that our attack is able to recover some training images but with limited quality. Future work could look at improving these attacks.

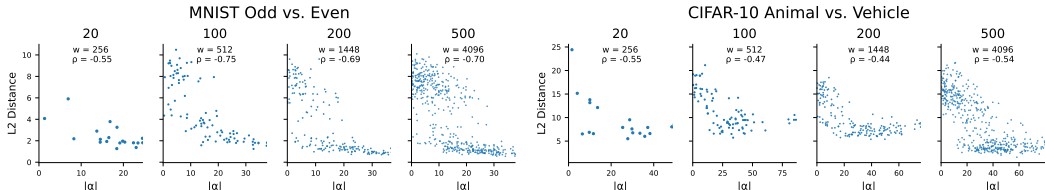

Figure 7: Scatter plots of reconstruction quality measured in l2 distance and corresponding $|\alpha|$ values for images. We vary the width with $n$ so that we observe a range of reconstruction qualities. $|\alpha|$ is negatively correlated with reconstruction error.

## 6    WHAT DATAPOINTS ARE SUSCEPTIBLE TO RECONSTRUCTION?

It has been observed in previous work that no datapoints are equally susceptible to privacy attacks (Carlini et al., 2022; Feldman & Zhang, 2020a; Carlini et al., 2021; Bagdasaryan & Shmatikov, 2019). In particular, *outlier* images tend to be leaked more easily than others. In this section, we show that this occurs for our attack, and provide theoretical justification for this.

### 6.1    Hard to fit Implies Easy to Reconstruct

By considering our reconstruction loss $\left\| \Delta\theta - \sum_j \alpha_j \phi(x_j) \right\|_2^2$ with $\phi(x_j) = \nabla_{\theta_f} f_{\theta_f}(x_j)$ we aim to learn a basis to "explain" $\Delta\theta$, we see that this could be cast as a sparse coding problem. Assuming that all $\phi(x_j)$ are of roughly the same magnitude, we expect the the parameters which larger $\alpha$ parameters to contribute more to $\Delta\theta$, and thus be more easily reconstructed. This is closely related to how data points with high influence are likely to be memorized (Feldman & Zhang, 2020a). We verify this heuristic holds empirically by plotting a scatter plot of reconstruction error vs. the corresponding $|\alpha|$ values calculated for infinite width in fig. 7. We see that images with small $\alpha$ val-

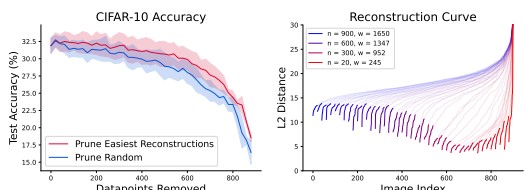

Figure 6: Test accuracy of iteratively pruned CIFAR-10 using either random pruning or pruning based on easily reconstructed images (Left), and reconstruction curves for pruned CIFAR-10 (Right). We see that easily reconstructed datapoints can be removed without harming accuracy. We observe a privacy "onion" effect where removing easily reconstructed images reveals other images which are easy to reconstruct. Bolded lines indicate the 20 images removed after pruning.

ues are "protected" from reconstruction since their contribution to $\Delta\theta$ is small and could be written off as noise. From appendix E, we know that $\alpha = 0$ corresponds to an image/label which does not alter the prediction at all, so this suggests that well-predicted datapoints are safe from reconstruction. Aside from the definition of $\alpha = K^{-1}y$, we can alternatively show that $\alpha$ is closely related to how *quickly* the model fits that datapoint. We can write (see appendix E.1 for a derivation) that $\alpha_j = \int_0^\infty (y_j - f_{\theta_t}(x_j))dt$ implying that datapoints which are slow to fit will have large $\alpha$ values, further strengthening the claim that outliers are easier to reconstruct.

### 6.2    A Reconstruction Privacy Onion

Carlini et al. (2022) showed that removing "vulnerable" training points reveals another set of training points which are susceptible to inference attacks. Here, we verify that our reconstruction attacks sees a similar phenomenon. Specifically, we consider reconstructing CIFAR-10 training images after training on $n$ datapoints, with $n = 900$ initially. We train and attack networks with $w \propto \sqrt{n}$, and then iterative remove the 20 most easily reconstructed datapoints based on the reconstruction curve. We scale the network capacity with $n$ so that our attack is unable to reconstruct the entire training set. We see that in fig. 6, that despite our network and attack capacity decreasing as we remove more datapoints, we are still able to reconstruct data points with increasing attack quality, replicating the "privacy onion" effect. Future work could look at how the interaction of $\alpha$ parameters affects which items are susceptible to reconstruction post-datapoint removal. Likewise, we evaluate the test accuracy on these pruned subsets in fig. 6. We see that as, these easily reconstructed datapoints tend to be outliers, removing them has a reduced effect on the test accuracy, however as Sorscher et al. (2022) discusses, the decision to remove easy vs. hard datapoints during pruning is dependent on other factors such as the size of the dataset and the complexity of the task.

## 7 UNIFYING RECONSTRUCTION AND DISTILLATION

In the previous sections, we considered the task of reconstructing the *entire* training set. To do this, we set the reconstruction image count $M > N$. What happens if we set $M < N$? Do we reconstruct a subset of a few training images, or do we form images that are *averages* of the training set?

We perform this experiment on the CIFAR-10 Animal/Vehicle task with 500 training images for a 4096-width model with linearized dynamics, aiming to reconstruct only 20 images. We recover the images shown in appendix G. With a few exceptions, these images are not items of the training set, but rather these images look like *averages* of classes.

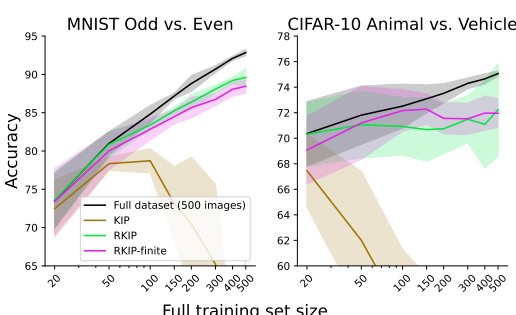

Now, what if we retrain a network on these reconstructed images? Noting that the optimal set of dual parameters for a full reconstruction is given by $\alpha^T = K_{TT}^{-1} y_T$, a natural choice for the training labels for these images is $y_R = K_{RR}\alpha^R$, where we compute the empirical NTK for $K_{RR}$ and use the learned $\alpha^R$

Figure 8: Performance of KIP, RKIP, and RKIP-finite on distill N images down to 20 images, trained on a 4096 width network with standard dynamics. KIP fails to transfer to finite networks while RKIP variations succeed.

parameters found during reconstruction. Retraining a different network from scratch on these 20 recovered images yields high accuracy, as shown in table 1 as RKIP-finite. This suggests that by doing this reconstruction we performed *dataset distillation*, that is we constructed a smaller set of images that accurately approximates the full dataset. This is not a coincidence, and the two algorithms are in fact the same. More formally:

**Theorem 2.** *The infinite-width variant reconstruction scheme of Eq. 4 with KKT points of Eq. 5 and Eq. 6, with $M \leq N$ where $M$ is the reconstruction image counts and $N$ is the dataset size, can be written as a kernel inducing point distillation loss under a different norm plus a non-negative correction term $\lambda$ as follows:*

$$\mathcal{L}_{Reconstruction} = \overbrace{\|y_T - K_{TR}K_{RR}^{-1}y_R\|_{K_{TT}^{-1}}^2 + \lambda_{var\ of\ R|T}}^{RKIP\ loss}$$

The full proof is given in appendix F. $\lambda_{\text{var of } R|T}$ is proportional to the variance of the reconstruction data points conditioned on the training data, based on the NTK (see appendix F). Intuitively, it ensures that training images provide "information" about the reconstructions. Compared to the loss of a well-known dataset-distillation algorithm, KIP (with $S$ referring to the distilled dataset):

$$\mathcal{L}_{\text{KIP}} = \|y_T - K_{TS}K_{SS}^{-1}y_S\|_2^2$$

The connection is apparent: the reconstruction loss is equal to the KIP dataset distillation loss under a different norm, where, rather than weighting each datapoint equally, we weight training images by their inverse similarity measured by the NTK, plus $\lambda_{\text{var of } r|T}$. This leads to a variant of KIP which we call Recon-KIP (RKIP) which uses the reconstruction loss in theorem 2. Note that for large datasets, RKIP is not practically feasible since it requires computing $K_{TT}^{-1}$, which is typical $N \times N$. We deal with small datasets in this work so it is still tractable.

We summarize the performance on KIP and RKIP in table 1 on the MNIST Odd/Even and CIFAR-10 Animal/Vehicle task, distilling 500 images down to 20. We evaluate 4096-width networks with standard or linearized dynamics, and infinite width using the NTK. Additionally, we consider using the images/labels made from reconstructing dataset points using a finite network trained on the full dataset and call this RKIP-finite. Note in this case the labels are not necessarily $\{+1, -1\}$, as $K_{0,RR}\alpha^R$ are not guaranteed to be one-hot labels. Similar results for distilling fewer images (20 - 500 training images) to 20 distilled images are shown in figure fig. 9.

We observe in table 1 that both KIP and RKIP have high infinite-width accuracies, but KIP sees a significant performance drop when transferring to finite networks. For example, while KIP achieves

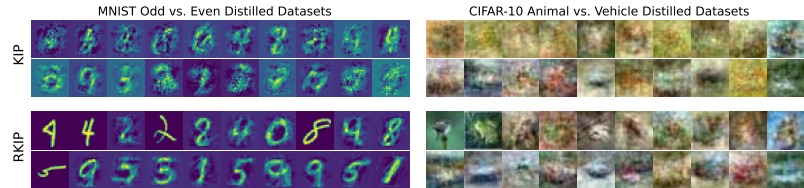

Figure 9: Visualizations of distilled datasets, on MNIST Odd/Even and CIFAR-10 Animal/Vehicle classification made with KIP, and RKIP. We distill datasets of 500 original images to 20 images (shown). KIP does not copy the original training images, while RKIP occasionally reproduces training images.

91.53% infinite-width test accuracy on the MNIST odd/even task, its finite-width performance is 55.62%, not significantly better than a random guess. Interestingly, this performance gap increases as we distill more images, as seen in fig. 8. For small distilled datasets there is little to no performance drop but for larger ones the difference is significant. In contrast, RKIP surprisingly sees almost no performance drop in the finite-width settings. We hypothesize that this finite-width transfer performance difference for KIP and not RKIP could be due to the contribution of $\lambda_{\text{var of } r|T}$, which we discuss in appendix F. We leave it to future work to explore this further. Additionally, RKIP-finite performs nearly as well as RKIP, despite distilling using the information from a single finite-width neural network.

## 8 Discussion, Limitations, and Conclusion

In this work we showed that a stronger variant of the attack given in Haim et al. (2022) which requires wide neural networks trained under MSE loss can provably reconstruct the entire training set, owing to the injectivity of the NTK kernel measure embedding. We showed that this attack works in practice for finite-width networks, with deviations from the infinite-width regime weakening the attack. We looked at how outlier datapoints are more likely to be reconstructed under our attack, and that these easily reconstructed images can be detrimental to learning. Finally, we made a novel connection between this reconstruction attack and dataset distillation. While this sheds light on dataset reconstruction attacks, and their theoretical underpinnings, there are still many avenues to explore.

In this work, we primarily explored 2-layer fully connected networks, where neural networks are known to behave similarly to their infinite-width counterparts. Meanwhile, deeper convolutional networks are known to deviate significantly, and it is unclear how well the attacks in this paper would transfer to those settings, and what adjustments would need to be made. Secondly, while we observed increasing model width increases the network's "resolving capacity" (i.e. how many images it could reconstruct), future work could look at how this quantity arises from deviations in the finite-width NTK from the infinite width one. Finally, we still need to resolve how the dataset reconstruction notion of privacy connects with more established notions such as differential privacy, which is the subject of future work.

We believe this work provides an important step toward understanding the strengths and weaknesses of dataset reconstruction attacks and provide novel connections to existing literature such as the Neural Tangent Kernel and dataset distillation.

## 8 Reproducbility Statement

This work uses open source datasets and models. Experimental details such as hyperparameters are described in appendix J. We additionally provide code for running the experiments in the supplementary material.

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

APPENDIX

# A    COMPARISON TO HAIM ET AL. (2022)

In section 3, we mentioned that we had problems reproducing Haim et al. (2022)'s attack. Here, we compare the two attacks and discuss the issues we found with theirs.

Firstly, we compare the quality of the two attacks. Haim et al. (2022) open sourced their code, as well as gave the best two sets of reconstructions for both CIFAR-10 and MNIST-10. We plot the reconstruction curves of these reconstructions here:

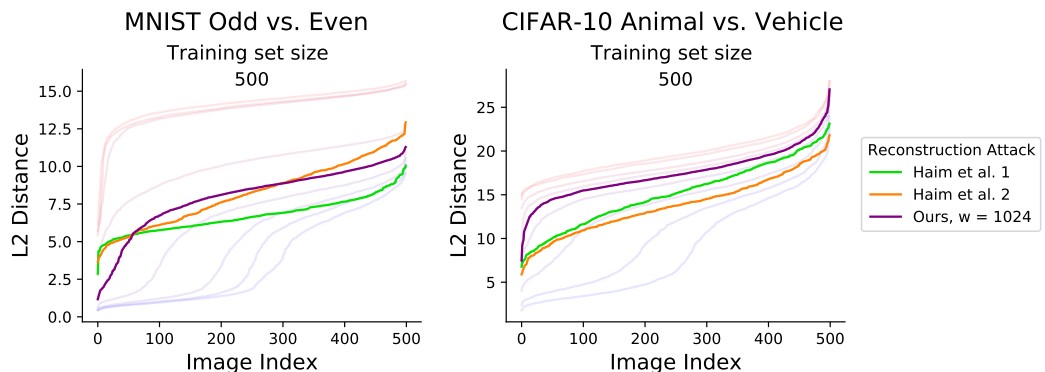

Figure 10: Reconstruction curves for the attacks given in Haim et al. (2022), in comparison to our reconstruction attacks, with a comparable width of 1024.

In their paper, they consider networks of a width 1000, so for a fair comparison, we highlight our reconstruction with a comparable width of 1024, under standard dynamics. We see that for MNIST, our attack has significantly better quality until image index 50 in which case both attacks perform poorly. For CIFAR-10, our attack performs worse but can achieve better performance with wider width or linearization.

Note that the two reconstruction curves presented from Haim et al. (2022) correspond to the two *best* reconstructions with carefully chosen hyperparameters. These hyperparameters are chosen to maximize reconstruction quality, which requires access to the training data to measure the quality. A priori, an attacker would not be able to assess reconstruction quality, and thus would not be able to do such hyperparameter tuning. In contrast, attack parameters such as learning rate/initialization were not fine-tuned for ours, and we use the same hyperparameters for every attack. Further gains could likely be seen with more careful tuning. We would like to emphasize that the goal of this work is not necessarily to create the strongest attack, but more so to explore the properties of the attack, and conditions for failure/success.

A further limitation of their attack is that it requires homogenous neural networks (i.e. no biases), in comparison to ours which uses biases, which is closer to practice. The largest limitation of their attack is that they require **a careful initialization scheme**, in which the weights of the first layer are initialized with significantly smaller variance. In Haim et al. (2022) they discuss that this is **essential** to the success of their attack. In contrast, we use the default initialization given in the Flax neural network library (Heek et al., 2020).

We also observed that the KKT condition given in eq. (1), which is required for their attack to work often is not reached in practice. To reiterate the definition of directional convergence, it requires that $\lim_{t\to\infty} \frac{\theta}{\|\theta\|_2} \to \frac{\theta'}{\|\theta'\|_2}$ for trained network parameters $\theta$, and $\theta'$ the solution to eq. (1). It is clear that if $\frac{\|\theta-\theta_0\|_2^2}{\|\theta_0\|_2^2} < 1$, i.e. the parameters have not drifted far from their initialization, then, we cannot hope the KKT point is reached (of course, with unlikely exceptions such as $\theta_0$ already being close to the KKT point). In practice, we found that $\|\theta_0\|_2^2 \approx 333$ and $\|\theta\|_2^2 \approx 377$, when running their attack, which suggests that the initialization still contributes a significant amount to the final parameters, suggesting that the KKT was not reached. Of course, our attack is not without limitations as well,

the most notable being that we require network initialization. We leave it to future work to alleviate this requirement.

## B PROOF OF THEOREM 1

Here we give a formal statement of theorem 1 as well as its proof.

**Theorem 1.** *If $\mathcal{L}_{reconstruction} \to 0$ (from Eq. 7), as $w \to \infty$ then $X_R \to X_T$ in probability for training data and recostruction data $X_T$ and $X_R$, respectively, on the unit hypersphere, with $w$ the network width.*

*Proof.* Define $k_\theta(x, x') = \nabla_\theta f_\theta(x)^\mathsf{T} \nabla_\theta f_\theta(x')$, that is, the finite-width/empirical NTK function. We know as network width $w \to \infty$, $\Delta\theta = \sum_{\alpha_i, x_i \in \alpha^T, X_T} \alpha_i \nabla_{\theta_0} f_{\theta_0}(x_i)$, with $\alpha^T = K_{\theta_0, TT}^{-1} y_T$, with $X_T$ being the training set, $y_T$ the training labels, and $K_{\theta_0, TT}$ the finite-width NTK evaluated on the training set. Our attack then becomes:

$$\left\| \Delta\theta - \sum_{\alpha_j x_j \in \alpha^R, X_R} \alpha_j \nabla_{\theta_f} f_{\theta_f}(x_j) \right\|_2^2 = \left\| \sum_{\alpha_i, x_i \in \alpha^T, X_T} \alpha_i \nabla_{\theta_0} f(x_i) - \sum_{\alpha_j x_j \in \alpha^R, X_R} \alpha_j \nabla_{\theta_f} f_{\theta_f}(x_j) \right\|_2^2 \tag{8}$$

$$= \left\| \sum_{\alpha_i, x_i \in \alpha^T, X_T} \alpha_i k_{\theta_0}(x_i, \cdot) - \sum_{\alpha_j x_j \in \alpha^R, X_R} \alpha_j k_{\theta_f}(x_j, \cdot) \right\|_2^2 \tag{9}$$

$$\xrightarrow{w \to \infty} \left\| \sum_{\alpha_i, x_i \in \alpha^T, X_T} \alpha_i k_{NTK}(x_i, \cdot) - \sum_{\alpha_j x_j \in \alpha^R, X_R} \alpha_j k_{NTK}(x_j, \cdot) \right\|_2^2 \quad \text{i.p.} \tag{10}$$

With $T$ and $R$ referring to the training and reconstruction set, respectively. We can take limits because we have as $w \to \infty$ we have that both $k_{\theta_0}, k_{\theta_f} \to k_{NTK}$ in probability (Jacot et al., 2018; Lee et al., 2019). Furthermore, define

$$P_T = \sum_{\alpha_i, x_i \in \alpha^T, X_T} \alpha_i \delta(x_i), \qquad P_R = \sum_{\alpha_j, x_j \in \alpha^R, X_R} \alpha_j \delta(x_j)$$

as measures associated with our trained network and reconstruction, respectively, and $\mu_* = \int_\Omega k_{NTK}(x, \cdot) dP_*(x)$, with $\Omega = S^d$, with $d$ being the data dimension (Assuming data lies on the unit hypersphere). We can now write:

$$\left\| \sum_{\alpha_i, x_i \in \alpha^T, X_T} \alpha_i k_{NTK}(x_i, \cdot) - \sum_{\alpha_j x_j \in \alpha^R, X_R} \alpha_j k_{NTK}(x_j, \cdot) \right\|_2^2 \tag{11}$$

$$= \left\| \int_\Omega k_{NTK}(x, \cdot) dP_T(x) - \int_\Omega k_{NTK}(x, \cdot) dP_R(x) \right\|_2^2 \tag{12}$$

$$= \| \mu_T - \mu_R \|_{\mathcal{H}_{NTK}}^2 \tag{13}$$

$\mu_T$ and $\mu_R$ are now kernel embeddings of our trained network and reconstruction, respectively. Our reconstruction loss becomes: $\| \mu_T - \mu_R \|_{\mathcal{H}_{NTK}}^2$. This is the maximum-mean discrepancy (MMD) (Gretton et al., 2012). We note that $P_T, P_R$ are signed Borel measures (since $\alpha$ are finite and our reconstruction/training sets are on the unit sphere). The NTK is universal over the unit sphere (Jacot et al., 2018), implying that the map $\mu :$ {Family of signed Borel measures} $\to \mathcal{H}$ is injective (Spiperumbudur et al., 2011). This means that $MMD_{k_{NTK}} = 0$ implies $P_R = P_T$ a.e., i.e. $X_R = X_T$ provided $\alpha_i \neq 0$ (see appendix E). Now putting everything together, we have that $\mathcal{L}_{reconstruction} \to 0$, so $MMD_{k_{NTK}} \to 0$ i.p. Note that the MMD and $k_{NTK}$ are continuous in their inputs, which implies that $X_R \to X_T$ as well. $\square$

## C CONNECTION TO GRADIENT INVERSION ATTACKS

Gradient inversion attacks (Zhu et al., 2019; Hatamizadeh et al., 2022) are reconstruction attacks which reconstruct training images given the network parameters and gradients at a specific time step for a specific batch $B$: $\theta_t, \nabla_{\theta_t} \mathcal{L}_B$. These attacks work by minimizing difference in gradient between loss on reconstruction images and on the known batch gradient:

$$\mathcal{L}_{\text{grad inv}} = ||\nabla_{\theta_t}\mathcal{L}_B - \nabla_{\theta_t}\mathcal{L}_R||_2^2 \tag{14}$$

Where $\mathcal{L}_R$ is the loss on reconstruction images. Compared to the attack presented in this work, ours does not use require the gradient directly, but the total change in parameters, $\Delta\theta$. We can consider $\Delta\theta$ as the sum of gradients over all time steps. With this in mind, gradient inversion attacks could be considered a special case of our attack where only one gradient step is taken and $\Delta\theta = -\eta\nabla_{\theta_t}\mathcal{L}_B$ for some learning rate $\eta$, similar to an extreme version of early stopping. Our theoretical analysis in theorem 1 applies here as well (as we do not require convergence, see appendix B and appendix I), so we can guarantee that gradient inversion attacks can reconstruct images when $\theta_t$ is near NTK initialization. As discussed in section 4, neural networks, generally speaking, do not behave in this regime, so our analysis cannot be directly used to explain the efficacy of gradient inversion attacks.

## D  RECONSTRUCTION ATTACK ALGORITHM DETAILS

---

**Algorithm 1** Standard Reconstruction Attack

---

**Require:** Initial Parameters $\theta_0$, final parameters $\theta_f$, network function $f_\theta$, randomly initialized reconstruction images and dual parameters $\{X_R, \alpha_R\}$, optimizer `Optim(params, gradients,` number of steps $T$
$\quad \Delta\theta = \theta_f - \theta_0$
$\quad t \leftarrow 1$
$\quad$ **while** $t < T$ **do**
$\quad\quad G \leftarrow \sum_i \alpha_i \nabla_\theta f_{\theta_f}(x_i)$ for $\alpha_i, x_i \in \alpha_R, X_R$ $\qquad\qquad$ ▷ Compute reconstruction gradient
$\quad\quad \mathcal{L}_{\text{recon}} = ||\Delta\theta - G||_2^2$ $\qquad\qquad\qquad\qquad\qquad$ ▷ Compute reconstruction loss
$\quad\quad \alpha_R, X_R \leftarrow \text{Optim}\left(\{\alpha_R, X_R\}, \frac{\partial L_{recon}}{\partial\{\alpha_R, X_R\}}\right)$ $\qquad$ ▷ Update Reconstruction Images
$\quad\quad t \leftarrow t + 1$
$\quad$ **end while**

---

---

**Algorithm 2** Batched Reconstruction Attack

---

**Require:** Initial Parameters $\theta_0$, final parameters $\theta_f$, network function $f_\theta$, white noise initialized reconstruction images and dual parameters $\{X_R, \alpha_R\}$, optimizer `Optim(params, gradients,` number of steps $T$, batch size $|B|$
$\quad \Delta\theta = \theta_f - \theta_0$
$\quad G_R = \sum_i \alpha_i \nabla_\theta f_{\theta_f}(x_i)$ for $\alpha_i, x_i \in \alpha_R, X_R$ $\quad$ ▷ Compute total reconstruction gradient (this step can also be batched)
$\quad$ **while** $t < T$ **do**
$\quad\quad$ Sample batch $\alpha_B, X_B \subset \alpha_R, X_R$ of batch size $|B|$ uniformly
$\quad\quad G_B \leftarrow \sum_i \alpha_i \nabla_\theta f_{\theta_f}(x_i)$ for $\alpha_i, x_i \in \alpha_B, X_B$ $\quad$ ▷ Compute reconstruction gradient for batch
$\quad\quad G_{B,old} \leftarrow \text{detach}(G_B)$ $\qquad\qquad\qquad\qquad\qquad$ ▷ Store old batch gradient
$\quad\quad \mathcal{L}_{\text{recon}} \leftarrow ||\Delta\theta - (G_R - G_{B,old} + G_B)||_2^2$ $\qquad$ ▷ Compute reconstruction loss
$\quad\quad \alpha_B, X_B \leftarrow \text{Optim}(\{\alpha_B, X_B\}, \partial L_{recon}/\partial\{\alpha_B, X_B\})$ $\qquad$ ▷ Optimize batch images
$\quad\quad G_{B,new} \leftarrow \sum_i \alpha_i \nabla_\theta f_{\theta_f}(x_i)$ for $\alpha_i, x_i \in \alpha_B, X_B$ $\qquad$ ▷ Compute new batch gradient
$\quad\quad G_R \leftarrow G_R - G_{B,old} + \text{detach}(G_{B,new})$ $\qquad$ ▷ Update total reconstruction gradient
$\quad\quad t \leftarrow t + 1$
$\quad$ **end while**

---

Here we discuss the runtime of our attack, given a model with $P$ parameters, $M$ reconstruction images, and $T$ iterations. We present two versions of the attack: the standard version of the attack, given in algorithm 1, and a minibatched version of the attack algorithm 2. Both versions of the attack are mathetmatically equivalent, but the batched version allows for larger reconstruction sets that may not fit into memory all at once.

For the standard version of the attack presented in algorithm 1, $O(MPT)$ time is required and $O(MP)$ memory is required, as naively one needs to store (and backpropagate through) gradients for each of the reconstructed images. Evidently for large datasets, one cannot pass the whole dataset

through the model at once and backpropagate, and as we need $M > N$ to reconstruct the full dataset, this seems problematic. For datasets in our paper, this was not a concern, but for larger datasets it would be.

To mitigate this issue, we present a minibatched version of the attack in algorithm 2, which requires $O(BPT)$ time and $O(BP)$ memory. The premise of this version that you store the a buffered value of the total sum of gradients $G_R = \sum_i^M g_i$ over reconstruction examples, and by carefully using autodiff, you can update only a subset of the gradients (using the buffered value of $G_R$ and subtracting the batch gradients). In algorithm 2, the `detach` function refers to the autodiff graph detachment function common in most autodiff libraries. This method has been implemented and performs exactly the same as the original attack, however the results in this paper do not require it as we dealt with small $N$.

# E  LABEL CONDITIONS FOR FULL RECOVERY

As discussed in section 3, we require that $\alpha_i \neq 0$ in order to recover the training image. Here we discuss the conditions for this to occur. We know that $\alpha = K^{-1}y$, and without loss of generality, consider reconstructing the final image at index $N$. Our equation for $\alpha$ becomes, focusing on the value of $\alpha_N$:

$$\begin{bmatrix} \alpha_{:N-1} \\ \alpha_N \end{bmatrix} = \begin{bmatrix} K_{:N-1,:N-1} & K_{:N-1,N} \\ K_{:N-1,N}^\intercal & K_{N,N} \end{bmatrix}^{-1} \begin{bmatrix} y_{:N-1} \\ y_N \end{bmatrix}$$

$$\begin{bmatrix} \alpha_{:N-1} \\ \alpha_N \end{bmatrix} = \begin{bmatrix} K_{:N-1,:N-1}^{-1} + K_{:N-1,:N-1}^{-1} K_{:N-1,N} Q K_{:N-1,N}^\intercal K_{:N-1,:N-1}^{-1} & -K_{:N-1,:N-1}^{-1} K_{:N-1,N} Q \\ -Q K_{:N-1,N}^{\intercal} K_{:N-1,:N-1}^{-1} & Q \end{bmatrix} \begin{bmatrix} y_{:N-1} \\ y_N \end{bmatrix}$$

With $Q = (K_{N,N} - K_{:N-1,N} K_{:N-1,:N-1}^{-1} K_{:N-1,N}^\intercal)^{-1}$. Setting $\alpha_N = 0$:

$$0 = -Q K_{:N-1,N}^T K_{:N-1,:N-1}^{-1} y_{:N-1} + Q y_N$$

$$y_N = K_{:N-1,N}^T K_{:N-1,:N-1}^{-1} y_{:N-1}$$

Noting $K_{:N-1,N}^T K_{:N-1,:N-1}^{-1} y_{:N-1}$ corresponds to the kernel regression prediction of $y_N$ given $y_{:N-1}$, we see that $\alpha_N = 0$ iff $x_N$ is already perfectly predicted by the remaining data points. Of course, the label corresponding exactly to the prediction occurs with probability 0.

### E.1 ALTERNATIVE INTERPRETATION OF $\alpha$

In section 6, we discussed how $\alpha$ parameters can be treated as items which are "hard" vs "easy" to fit. Here we derive how $\alpha_j = \int_0^\infty (y_j - f_{\theta_t}(x_j))dt$ for MSE loss (up to some scaling parameter).

$$\mathcal{L}(\theta_t) = \frac{1}{2}\sum_i (y_i - f_{\theta_t}(x_i))^2$$

$$\frac{\partial \theta_t}{\partial t} = -\eta \frac{\partial \mathcal{L}(\theta_t)}{\partial \theta_t}$$

$$\frac{\partial \theta_t}{\partial t} = -\eta \sum_i (y_i - f_{\theta_t}(x_i))\frac{\partial f_{\theta_t}(x_i)}{\partial \theta_t}$$

$$\Delta\theta = \int_0^\infty \frac{\partial \theta_t}{\partial t}dt$$

$$= \int_0^\infty -\eta \sum_i (y_i - f_{\theta_t}(x_i))\frac{\partial f_{\theta_t}(x_i)}{\partial \theta_t}dt$$

$$= -\eta \sum_i \left[ \int_0^\infty (y_i - f_{\theta_t}(x_i))\frac{\partial f_{\theta_t}(x_i)}{\partial \theta_t}dt \right]$$

$$\approx -\eta \sum_i \left[ \int_0^\infty (y_i - f_{\theta_t}(x_i))dt \frac{\partial f_{\theta_0}(x_i)}{\partial \theta_0} \right] \quad \text{(Frozen Kernel approximation)}$$

Noting that $\Delta\theta = \sum_i \alpha_i \frac{\partial f_{\theta_0}(x_i)}{\partial \theta_0}$ (eq. (5)) and matching terms, also noting that $P > T$ (more parameters than training points), we have the system is uniquely solved when $\alpha_i = -\eta \int_0^\infty (y_j - f_{\theta_t}(x_j))dt$. We can verify this experimentally, by plotting the calculated values of $\alpha_i$ vs. $\int_0^\infty (y_j - f_{\theta_t}(x_j))dt$ in fig. 11.

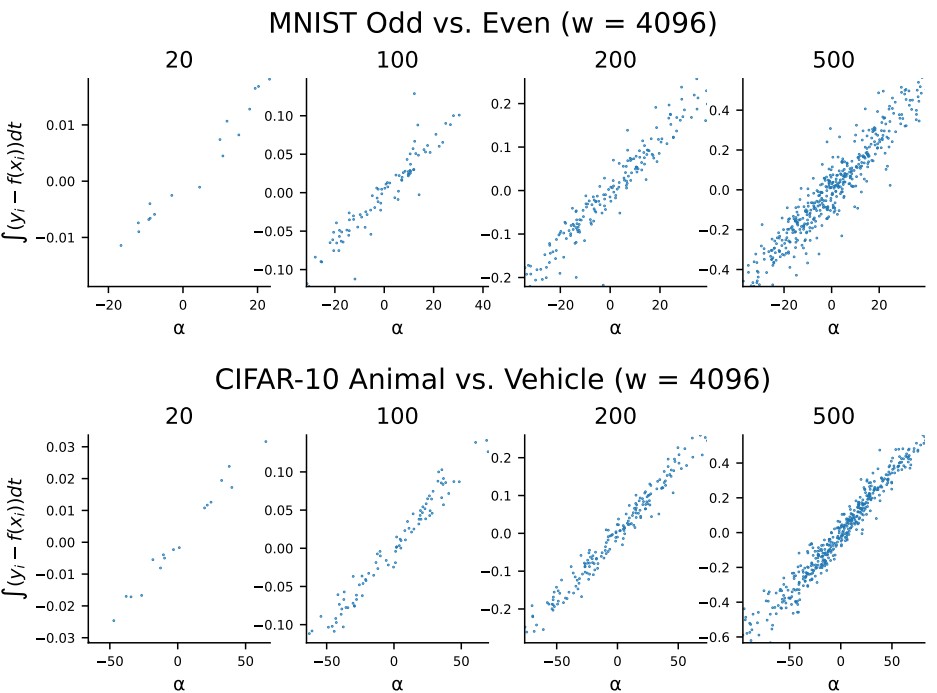

Figure 11: Infinite width values of $\alpha$ vs. error integral formulation of error calculated from 4096-width finite networks. There is a strong correlation between the two values.

# F PROOF OF THEOREM 2: RKIP DERIVATION FROM RECONSTRUCTION LOSS

*Proof.* Here we derive how one gets from the reconstruction loss given in eq. (7) to theorem 2, the RKIP loss. We repeat the losses here:

$$\mathcal{L}_{\text{Reconstruction}} = \|\Delta\theta - \alpha^{\mathsf{T}}\nabla_\theta f_{\theta_0}(X_T)\|_2^2 + \lambda_{\text{var of } R|T}$$

$$\mathcal{L}_{\text{RKIP}} = \|y_T - K_{TR}K_{RR}^{-1}y_R\|_{K_{TT}^{-1}}^2$$

First we note that $\alpha^T = K_{TT}^{-1}y_T$, and if we trained on reconstructions with labels $y_R$, then $\alpha^R = K_{RR}^{-1}y_R$. For brevity we denote $s_i^* = \{\alpha_i^*, x_i^*\}$ and $S^* = \{\alpha^*, X_*\}$. Let $S^T$ and $S^R$ denote the training and reconstruction set, respectively

$$\left\|\Delta\theta - \sum_{s_j^R \in S^R} \alpha_j^R \nabla_{\theta_f} f_{\theta_f}(x_j^R)\right\|_2^2$$

$$= \left\|\sum_{s_i^T \in S^T} \alpha_i^T \nabla_{\theta_0} f(x_i^T) - \sum_{s_j^R \in S_R} \alpha_j^R \nabla_{\theta_f} f_{\theta_f}(x_j^R)\right\|_2^2$$

$$= \left\|\sum_{s_i^T \in S^T} \alpha_i^T k_{\theta_0}(x_i^T, \cdot) - \sum_{s_j^R \in S_R} \alpha_j^R k_{\theta_f}(x_j^R, \cdot)\right\|_2^2$$

Again take the infinite width limit so $k_{\theta_0}, k_{\theta_f} \to k_{NTK}$, we which just write $k$ for simplicity.

$$\left\|\sum_{s_i^T \in S^T} \alpha_i^T k_{\theta_0}(x_i^T, \cdot) - \sum_{s_j^R \in S_R} \alpha_j^R k_{\theta_f}(x_j^R, \cdot)\right\|_2^2 \to \left\|\sum_{s_i^T \in S^T} \alpha_i^T k(x_i^T, \cdot) - \sum_{s_j^R \in S_R} \alpha_j^R k(x_j^R, \cdot)\right\|_2^2$$

$$= \sum_{s_i^T \in S^T}\sum_{s_j^T \in S^T} \alpha_i^T \alpha_j^T k(x_i^T, x_j^T) - 2\sum_{s_i^T \in S^T}\sum_{s_j^R \in S^R} \alpha_i^T \alpha_j^R k(x_i^T, x_j^R)$$

$$+ \sum_{s_i^R \in S^R}\sum_{s_j^R \in S^R} \alpha_i^R \alpha_j^R k(x_i^R, x_j^R)$$

$$= \alpha^{T\mathsf{T}} K_{TT} \alpha^T - 2\alpha^{T\mathsf{T}} K_{TR} \alpha^R + \alpha^{R\mathsf{T}} K_{RR} \alpha^R$$

$$= y_T^{\mathsf{T}} K_{TT}^{-1} K_{TT} K_{TT}^{-1} y_T - 2 y_T^{\mathsf{T}} K_{TT}^{-1} K_{TR} K_{RR}^{-1} y_R + y_R^{\mathsf{T}} K_{RR}^{-1} K_{RR} K_{RR}^{-1} y_R$$

$$= y_T^{\mathsf{T}} K_{TT}^{-1} y_T - 2 y_T^{\mathsf{T}} K_{TT}^{-1} K_{TR} K_{RR}^{-1} y_R + y_R^{\mathsf{T}} K_{RR}^{-1} y_R$$

$$= y_T^{\mathsf{T}} K_{TT}^{-1} y_T - 2 y_T^{\mathsf{T}} K_{TT}^{-1} K_{TR} K_{RR}^{-1} y_R + y_R^{\mathsf{T}} K_{RR}^{-1} K_{RT} K_{TT}^{-1} K_{TR} K_{RR}^{-1} y_R$$

$$- y_R^{\mathsf{T}} K_{RR}^{-1} K_{RT} K_{TT}^{-1} K_{TR} K_{RR}^{-1} y_R + y_R^{\mathsf{T}} K_{RR}^{-1} y_R$$

$$= \|y_T - K_{TR} K_{RR}^{-1} y_R\|_{K_{TT}^{-1}}^2 + y_R^{\mathsf{T}}(K_{RR}^{-1} - K_{RR}^{-1} K_{RT} K_{TT}^{-1} K_{TR} K_{RR}^{-1}) y_R$$

$$= \underbrace{\|y_T - K_{TR} K_{RR}^{-1} y_R\|_{K_{TT}^{-1}}^2}_{\text{label matching}} + \underbrace{y_R^{\mathsf{T}} K_{RR}^{-1}(K_{RR} - K_{RT} K_{TT}^{-1} K_{TR}) K_{RR}^{-1} y_R}_{\lambda_{\text{var of } R|T}}$$

$\square$

$\lambda_{\text{var of } R|T}$ is proportional to $K_{RR} - K_{RT} K_{TT}^{-1} K_{TR}$, which is $K_{[T,R],[T,R]}/K_{TT}$, the Schur complement of $K_{[T,R],[T,R]}$ with $K_{TT}$. Note that this is the Gaussian conditional variance formula, assuming we are making predictions of $R$ based on $T$. This regularizer ensures that not only do the reconstructed images result in the correct predictions (ensured by the label matching term) but also that our distilled dataset images do not deviate significantly from the training distribution, as measured by the NTK. We hypothesize this term is what contributes to the success of RKIP over KIP in the finite-width setting. As there is nothing that directly ensures that KIP distilled datapoints remain "similar" to training images (only that they are predictive), these distilled images may be more susceptible to domain shift, such as moving from the infinite-width setting to finite width. This interesting behavior could be the subject of future work.

## G    FINITE RKIP IMAGES

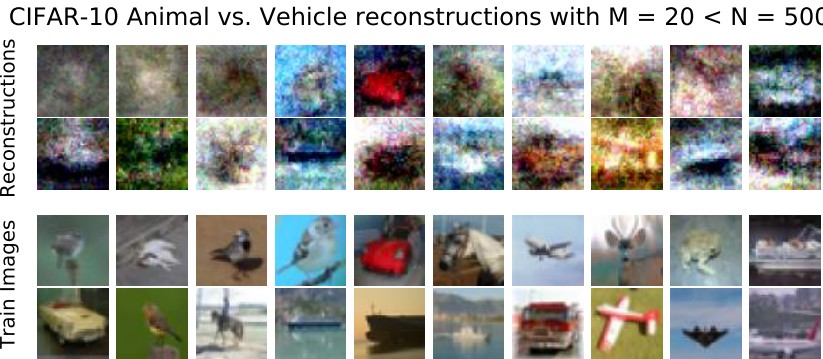

Figure 12: Reconstructing 20 images from a network trained on 500 CIFAR-10 images. Reconstructions often do not match actual training images and contain heavy corruption. Retraining on these images yields high accuracy.

fig. 13 and fig. 14 show the resulting reconstructions when reconstructing 20 images from a dataset that may contain up to 500 images. Reconstructions are made from 4096 networks with linearized dynamics.

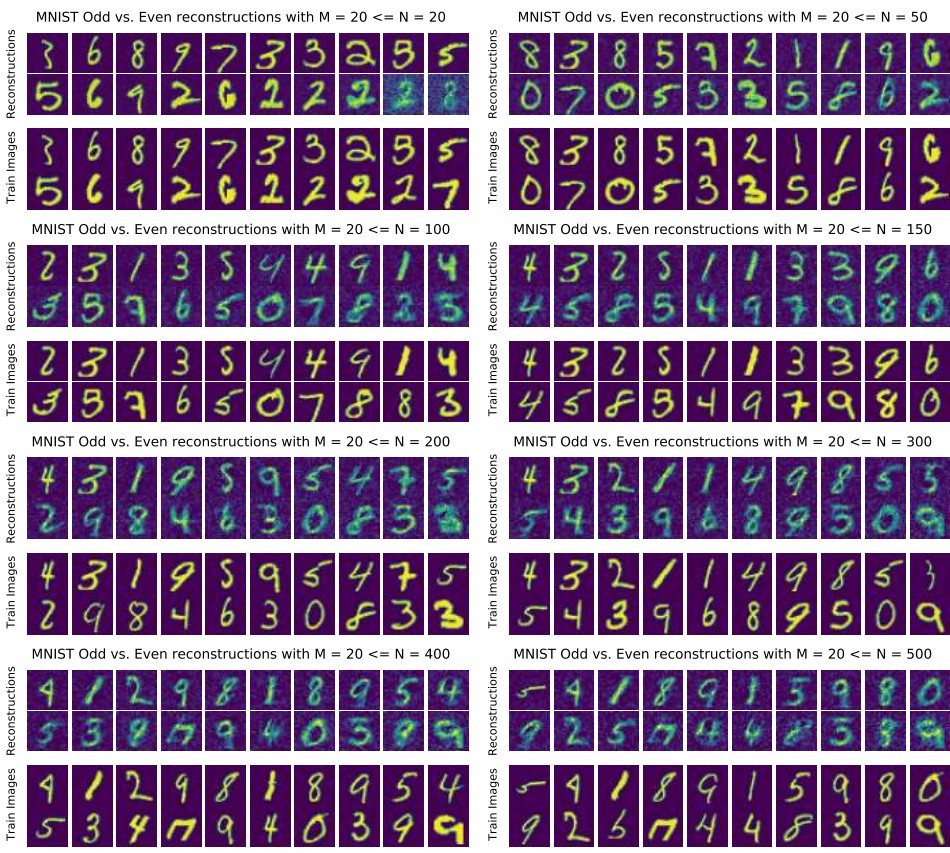

Figure 13: Reconstructing 20 images from a dataset that may be larger than 20 images (MNIST Odd vs. Even)

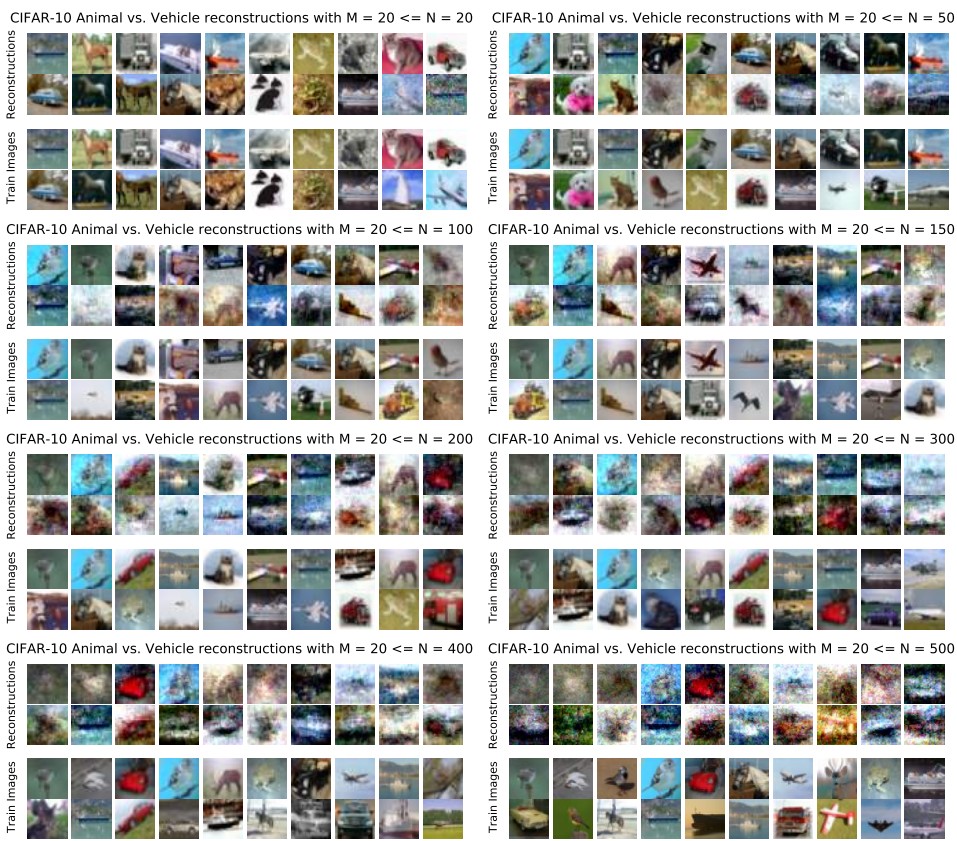

Figure 14: Reconstructing 20 images from a dataset that may be larger than 20 images (CIFAR-10 Animal vs. Vehicle)

## H  MULTICLASS RECONSTRUCTION LOSS

Here we derive the multi-class reconstruction loss which we use in section 4. Our least-norm predictor satisfies the following conditioned (assuming that the network behaves in the linear regime):

$$
\arg\min_{\Delta\theta} \frac{1}{2}\|\Delta\theta\|_2^2 \quad s.t.
$$
$$
\forall c \in [C], \quad \Delta\theta^\mathsf{T} \nabla_\theta f_{\theta_0}^c(X_T) = y_T^c - f_{\theta_0}^c(X_T). \tag{15}
$$

With $f_{\theta_0}^c$ referring to the network output on the $c$th class and $y_T^c$ referring to the training labels for the $c$th class. The network will converge to the least norm solution (norm of the difference from initialization), due to the network behaving in the lazy regime with gradient flow (de Azevedo , https://math.stackexchange.com/users/339790/rodrigo-de azevedo). Writing the equation with dual variables our full Lagrangian is

$$
\mathcal{L}(\Delta\theta, \alpha) = \frac{1}{2}\|\Delta\theta\|_2^2 + \sum_{c=1}^{C} \alpha^{c\mathsf{T}}\left(\Delta\theta^\mathsf{T}\nabla_\theta f_{\theta_0}^c(X_T) - (y_T^c - f_{\theta_0}^c(X_T))\right). \tag{16}
$$

With $\alpha$ our set of dual parameters $\in R^{C\times M}$, that is, we have a set of $M$ dual parameters for each class. Taking derives w.r.t $\Delta\theta$:

$$
0 = \nabla_{\Delta\theta}\mathcal{L}(\Delta\theta, \alpha) = \Delta\theta + \sum_{c=1}^{C} \alpha^{c\mathsf{T}}\left(\nabla_\theta f_{\theta_0}^c(X_T)\right). \tag{17}
$$

So our multiclass reconstruction loss is

$$\mathcal{L}_{\text{reconstruction}} = \left\| \Delta\theta - \sum_{c=1}^{C} \alpha^{c\mathsf{T}} \nabla_\theta f_{\theta_0}^c(X_T) \right\|_2^2. \tag{18}$$

We can use the same argument as in section 3 to show this attack is exact in the infinite width limit.

## I  EARLY STOPPING AND CROSS-ENTROPY

### I.1  EARLY STOPPING

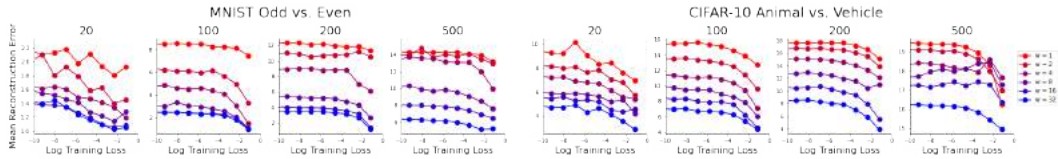

Figure 15: Mean reconstruction errors for networks trained to various early stopping losses from $10^{-1}$ to $10^{-6}$

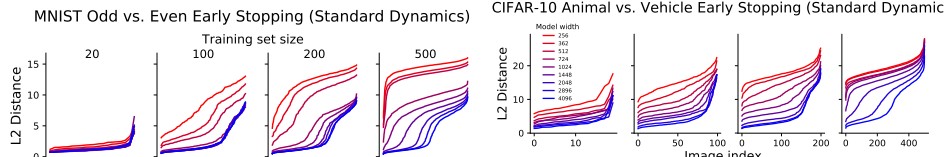

Figure 16: Reconstruction curves for networks trained to a loss of 1e-2, i.e. significant underfitting (under standard dynamics). Compared to fig. 2, we see that reconstruction quality is unaffected by early stopping, consistent with the theory.

In the main text, we considered models trained with low learning rates for $10^6$ epochs, so that we achieve the KKT conditions described in section 3. In practice, this is computationally expensive, and often ill-advised due to overfitting, with early stopping being a common regularization technique. One major limitation of the attack proposed in Haim et al. (2022) is that it requires the network to reach the KKT point to perform the attack. Our method **does not require the model to reach convergence** for the attack to work. Again we note that the time evolution of network parameters is given by:

$$\Delta\theta(t) = \nabla_\theta f_{\theta_0}(X_T)^{\mathsf{T}} \underbrace{K_0^{-1} \left( I - e^{-\eta K_0 t} \right) \left( y_T - f_{\theta_0}(X_T) \right)}_{\text{time-dependent weights, } \alpha(t)}$$

 Notably, even at finite time, the change in network parameters is still a linear combination of the finite-width NTK feature maps of the training set, $\nabla_\theta f_{\theta_0}(X_T)$, with the indicated *time dependent* weights, $\alpha(t)$. Note that the success of the attack in infinite width relies on the injectivity of the kernel measure embedding, not that $\alpha(t)$ be at its converged value, implying the attack works with early stopping. The caveat with the early stopping attack is that we cannot necessarily hope to recover the original training labels.

We verify that this attack works in practice by repeating the attack procedure in section 4, with early stopping. We apply the attack on networks that achieve a mean training loss of from $10^{-1}$ to $10^{-8}$ (note that this means that on for $\{+1, -1\}$ labels, the outputs were around $0.45$ off, i.e. quite underfit, in the case of $\mathcal{L} = 10^{-1}$), with results shown in fig. 15, with the specific reconstruction curve for $\mathcal{L} = 10^{-2}$ in fig. 16. We observe that early stopping in general *improves* reconstruction quality. We posit that there are two possible reasons for this: firstly, that early in training there is less time for the kernel to evolve, so the network exhibits network dynamics closer to the lazy regime. Secondly, we hypothesize that this could be that early in training all datapoints have a roughly equal

contribution to the parameter changes, whereas later in training, certain datapoints have a stronger influence (see section 6 and appendix E.1). When some datapoints have a much stronger influence than others, this could cause the the less influential datapoints to be "drowned out" by the signal of the more influential ones. Future work could study the complex relationship between early stopping, outlier influence, and network vulnerability more closely. We also note that in the limit of a single gradient step, our attack corresponds to *gradient leakage* attacks (Zhu et al., 2019).

## I.2 CROSS ENTROPY LOSS

Following (Lee et al., 2019), we have that wide networks trained under cross-entropy loss also exhibit lazy dynamics provided they are of sufficient width. The corresponding ODE is:

$$\mathcal{L}(\theta_t) = -\sum_i y_i \log \sigma(f_{\theta_t}(x_i))$$

$$\frac{\partial \theta_t}{\partial t} = -\eta \frac{\partial \mathcal{L}(\theta_t)}{\partial \theta_t}$$

$$\frac{\partial \theta_t}{\partial t} = -\eta \sum_i (\sigma(f_{\theta_t}(x_i)) - y_i) \frac{\partial f_{\theta_t}(x_i)}{\partial \theta_t}$$

Unlike eq. (3), there is not a closed form solution to this, however the key point is that $\Delta\theta$ still is a linear combination of $\nabla_\theta f_\theta(x_i)$.

## J EXPERIMENTAL DETAILS

### J.1 LIBRARIES AND HARDWARE

We use the JAX, Optax, Flax, and neural-tangents libraries (Bradbury et al., 2018; Babuschkin et al., 2020; Heek et al., 2020; Novak et al., 2020; 2022). All experiments were run on Nvidia Titan RTX graphics cards with 24Gb VRAM.

### J.2 NETWORK TRAINING

Unless otherwise stated, networks trained on real data are trained for $10^6$ iterations of full batch gradient descent, with SGD with momentum 0.9. For the learning rate, we set $\eta = N \times 2\mathrm{e}{-7}$, where $N$ is the number of training images. For distilled data, we use a learning rate of $\eta = N \times 6\mathrm{e}{-6}$, where $N$ is now the distilled dataset size. We did not find that results were heavily dependent on the learning rates used during training. Additionally, if the training loss was less than $1\mathrm{e}{-10}$, we terminated training early. Every reconstruction curve in the main text is the average of 3 unique networks trained on 3 unique splits of training data.

For binary classification, we use labels in $\{+1, -2\}$, and for 10-way multiclass classification, we use labels of 0.9 corresponding to the selected class and -0.1 for other classes.

### J.3 RECONSTRUCTIONS

To create reconstructions, we initialize reconstruction images with a standard deviation of 0.2, and dual parameters to be uniform random within $[-0.5, 0.5]$. We use Adam optimizer (Kingma & Ba, 2015), with a learning rate of 0.02 for all reconstructions. As stated in appendix A, these could be fine-tuned to improve performance. We optimize the images for 80k iterations. Like with Haim et al. (2022), we found that it was useful to use a `softplus` rather than a `Relu` during reconstruction, owing to the smoothing gradient loss. We annealed `softplus` temperature from 10 to 200 over the course of training, so that we are effectively using `ReLUs` by the end of training. Unless otherwise stated, we aim to reconstruct $M = 2N$ reconstruction images, with $N$ the training set size.

### J.4 DISTILLATION

We initialize distilled images with a standard deviation of 0.2 and distill for 50k iterations with Adam optimizer with a learning rate of 0.001.

### J.5 FINE-TUNING EXPERIMENTS

For the fine-tuning experiments in section 5, we use the pretrained ResNet-18 from flaxmodels (Wright, 2022). For these experiments, we used 64-bit training, as we found it improved reconstruction quality. We train for 30000 iterations with a learning rate of $1e - 5$. We use SGD with no momentum, and also freeze the batchnorm layers to the parameters used by the initial model. We use the hybrid loss described in appendix K.

### J.6 PRUNING EXPERIMENTS

For the pruning experiments we train for $10^5$ epochs for each training run with a learning rate of $n \times 3e - 6$ with SGD with momentum 0.9. For reconstruction we use the same reconstruction attack in the main text for 30000 iterations. For each training iteration we set with width of the network to be $w = 55\sqrt{n}$, as we found that generally kept the strength of the attack the same for different values of $n$. If we make the attack too strong, then too many training points will be reconstructed and we will no longer have any notion of "easy to reconstruct" examples, as all examples will be reconstructed equally well.

### J.7 RECONSTRUCTION POST-PROCESSING

We do not post-process our reconstructions and the reconstruction curves and visualization are based on unmodified reconstructions. Note that Haim et al. (2022) has a more complex reconstruction scheme involving rescaling the reconstructions and averaging, which is detailed in Haim et al. (2022)'s appendix.

## K CHOICE OF KERNEL

During reconstruction, there is a small choice we can make, specifically whether we using the initialization gradient tangent space, $\nabla_{\theta_0} f_{\theta_0}(x_i)$ or the final gradient tangent vector space, $\nabla_{\theta_f} f_{\theta_f}(x_i)$. Specifically, we can choose to optimize $\mathcal{L}_f$ or $\mathcal{L}_0$:

$$\mathcal{L}_f = \left\| \Delta\theta - \sum_{s_j^R \in S^R} \alpha_j^R \nabla_{\theta_f} f_{\theta_f}(x_j^R) \right\|_2^2$$

$$\mathcal{L}_0 = \left\| \Delta\theta - \sum_{s_j^R \in S^R} \alpha_j^R \nabla_{\theta_0} f_{\theta_0}(x_j^R) \right\|_2^2$$

Under linearized dynamics, these two are equal as $\nabla_\theta f_\theta(x)$ does not change. However for standard dyanmcis there is a small difference, with the difference increasing the more the kernel changes. For the results in the main text, we use $\mathcal{L}_f$. We found that this has 6marginally better performance than $\mathcal{L}_0$, but this difference is rather minor. This also leads to a third choice of reconstruction loss, which we call the "hybrid" loss $\mathcal{L}_h$, which considers the change in parameters a mixture of both the final and initial kernel:

$$\mathcal{L}_h = \left\| \Delta\theta - \sum_{s_j^R \in S^R} \alpha_{j,0}^R \nabla_{\theta_f} f_{\theta_f}(x_j^R) - \sum_{s_j^R \in S^R} \alpha_{j,f}^R \nabla_{\theta_0} f_{\theta_0}(x_j^R) \right\|_2^2$$

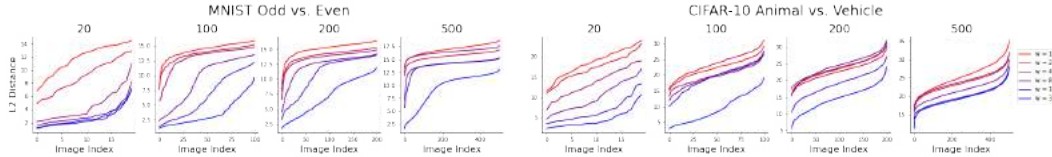

Figure 17: Reconstruction curves for convolutional architectures trained on MNIST Odd vs. Even and CIFAR-10 Animal vs. Vehicle classification with varying width multipliers from $1 - 32$

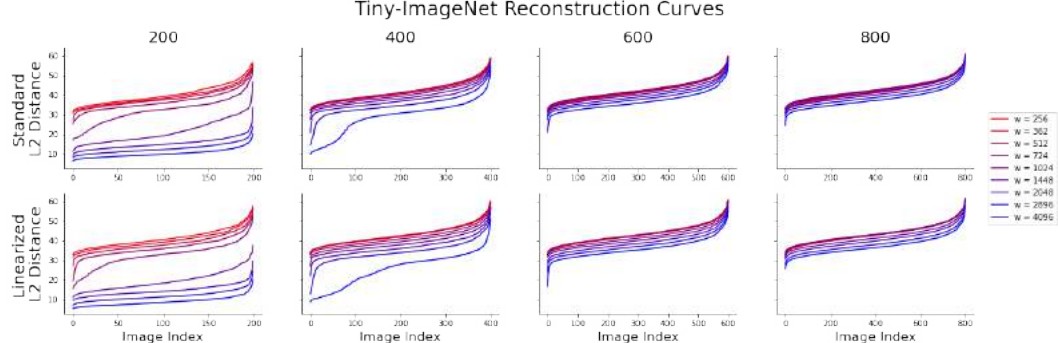

Figure 18: Reconstruction curves for networks trained on Tiny-ImageNet 200-way classification

In which we have two sets of dual parameters $\alpha_0$ and $\alpha_f$. This further increases performance, but in general is twice as slow as optimizing $\mathcal{L}_0$ or $\mathcal{L}_f$, and we chose to use this loss for the ResNet-18 fine-tuning experiments, since that setting is more challenging. Note that this attack could be generalized to include multiple checkpoints $\theta_t$ along the trajectory and more dual parameters, however of course this would require access to more training checkpoints.

## L    ADDITIONAL RESULTS

### L.1    CONVOLUTIONAL ARCHITECTURES

Here we provide results for our attack applied to convolutional architectures. We trained networks on binary MNIST and CIFAR-10 classifications tasks, as we did in the main text, but trained on a convolutional architecture from scratch. We use the common LeNet-5 architecture with width multipliers ranging from 1-32. fig. 17 shows the results. The findings observed on two-layer networks still apply in this settings, however our attack struggles more in deeper architectures.

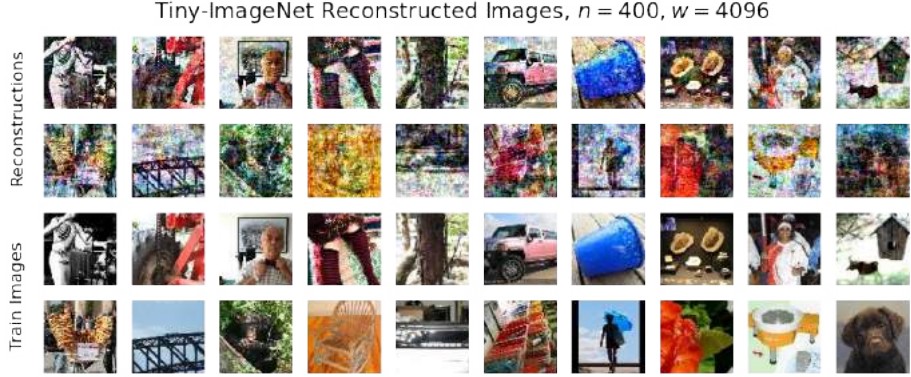

Figure 19: Reconstructed images and their nearest train iamge counterparts for Tiny-ImageNet reconstruction for a $w = 4096$ network with $n = 400$

## L.2    TINY-IMAGENET

The main text works mainly with classification on small, low-resolution datasets such as MNIST and CIFAR-10. Here we consider more complex datasets with higher resolution by applying our attack to Tiny-ImageNet classification. Tiny-ImageNet consists of 200 classes with images of resolution $64 \times 64$ (Le & Yang, 2015). As the quality of the NTK approximation is negatively affected by image resolution (Jacot et al., 2018), this experiment serves as an important testing ground of the viability of the attack for higher resolution images. We show the results on few-shot classification, consider $1 - 4$ images per classes in fig. 18 and with the reconstructed images in fig. 19. Reconstructing higher resolution images is more challenging and improving this attack on high resolution images is an interesting direction for future work.

## L.3    KERNEL DISTANCE VS. RECONSTRUCTION QUALITY SCATTER PLOTS FOR MULTICLASS CLASSIFICATIONS

fig. 20 shows the corresponding fig. 3 for multiclass classification. As we observed, multiclass classification has improved reconstruction quality. From fig. 20, we see that multiclass classification sees significantly lower kernel distances (up to 1e-2 for MNIST and 2e-2 for CIFAR-10) compared to binary classification (up to 3e-2 for MNIST and 9e-2 for CIFAR-10, see fig. 3), which may explain why the reconstructions have better quality.

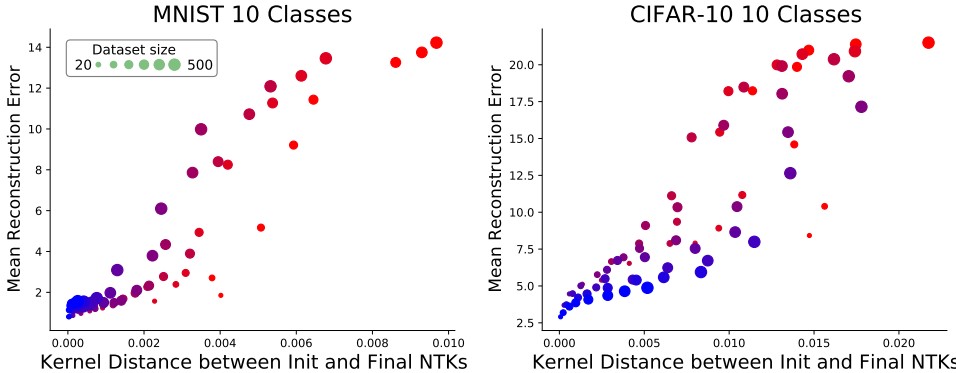

Figure 20: Mean reconstruction error vs. the kernel distance from the initialization kernel to the final kernel for multiclass classification. The mean reconstruction error, measured as the average value of the reconstruction curve, is strongly correlated with how much the finite-width NTK evolves over training. Dataset size is given by dot size, while the color indicates model width. Multiclass classification sees significantly lower kernel distances (up to 1e-2 for MNIST and 2e-2 for CIFAR-10) compared to binary classification (up to 3e-2 for MNIST and 9e-2 for CIFAR-10, see fig. 3), which may be the cause of better reconstruction quality.

## L.4    EXTRA PRUNING EXPERIMENTS

In section 6, we considered removed class balanced subsets of the training data at each training iteration, either by random or by ease of reconstruction. If instead we allow class imbalance, we see the same behaviour as in section 6 initially, but as more datapoints are removed, we see in fig. 21 that removing easy reconstructions results in strongly imbalanced classes, resulting in poor test accuracy. Understanding why some classes are more susceptible to reconstruction is likely related to the discussion in section 6. Additionally, we found that if we underfit the data, then we do not observe any difference in test accuracy for pruning random vs. reconstructions. This suggests that the effect of large $\alpha$ values only shows later in training, when some datapoints are well fit and others still underfit.

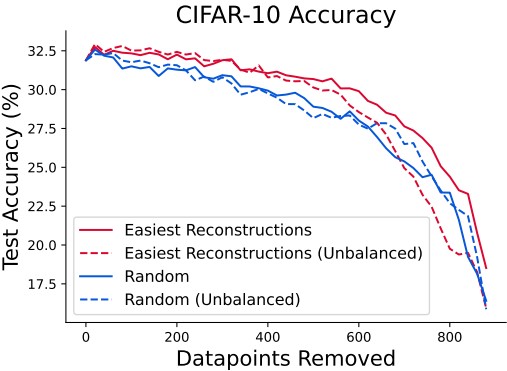

Figure 21: Test accuracy of iteratively pruned CIFAR-10 using either random pruning or pruning based on easily reconstructed images with either class balanced subsets or non-balanced subsets.

## L.5    ADDITIONAL RECONSTRUCTION CURVES

We show additional reconstruction curves for all dataset sizes in $[20, 50, 100, 150, 200, 300, 400, 500]$ for MNIST Odd vs. Even and CIFAR-10 Animal vs. Vehicle in fig. 22. We show the same reconstruction curves with the distillation reconstructions in **??**. fig. 23 shows the same reconstruction curves for early stopping. Finally, fig. 24 shows the same curves for multi-class classification.

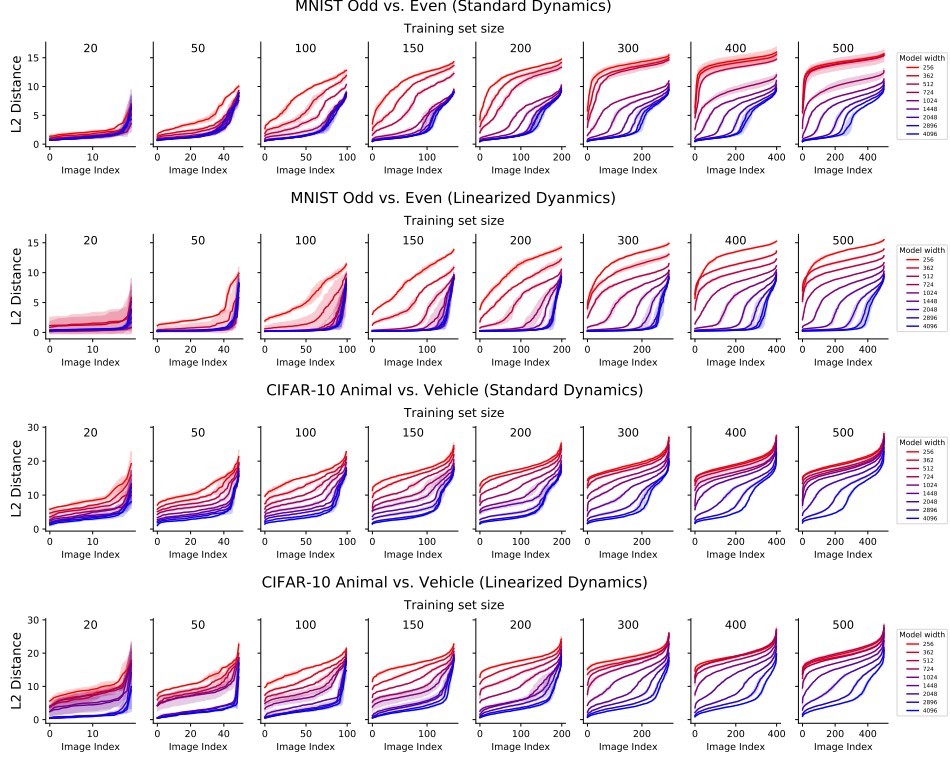

Figure 22: Reconstruction curves for binary classification tasks

## L.6    RECONSTRUCTION IMAGES

Here we show all the reconstruction images and their nearest training images in terms of $L_2$ distance. Images are sorted based on their rank in the reconstruction curve.

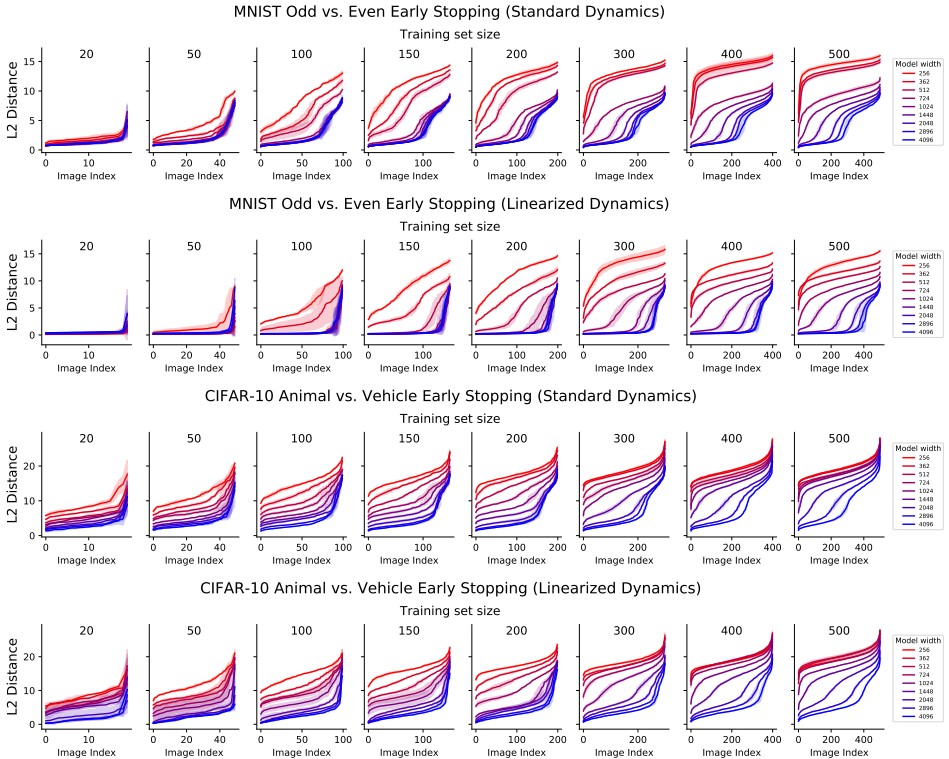

Figure 23: Reconstruction curves for binary classification tasks with early stopping

### L.6.1 BINARY CLASSIFICATION

We show the reconstruction curves for MNIST Odd vs. Even and CIFAR-10 Animal vs. Vehicle tasks for width 4096 and 1024 networks with linearized or standard dynamics in figures 25 to 32.

### L.6.2 MULTICLASS CLASSIFICATION

We show the reconstruction curves for MNIST and CIFAR-10 10-way classification for width 4096 and 1024 networks with linearized or standard dyanmics in figures 33 to 40.

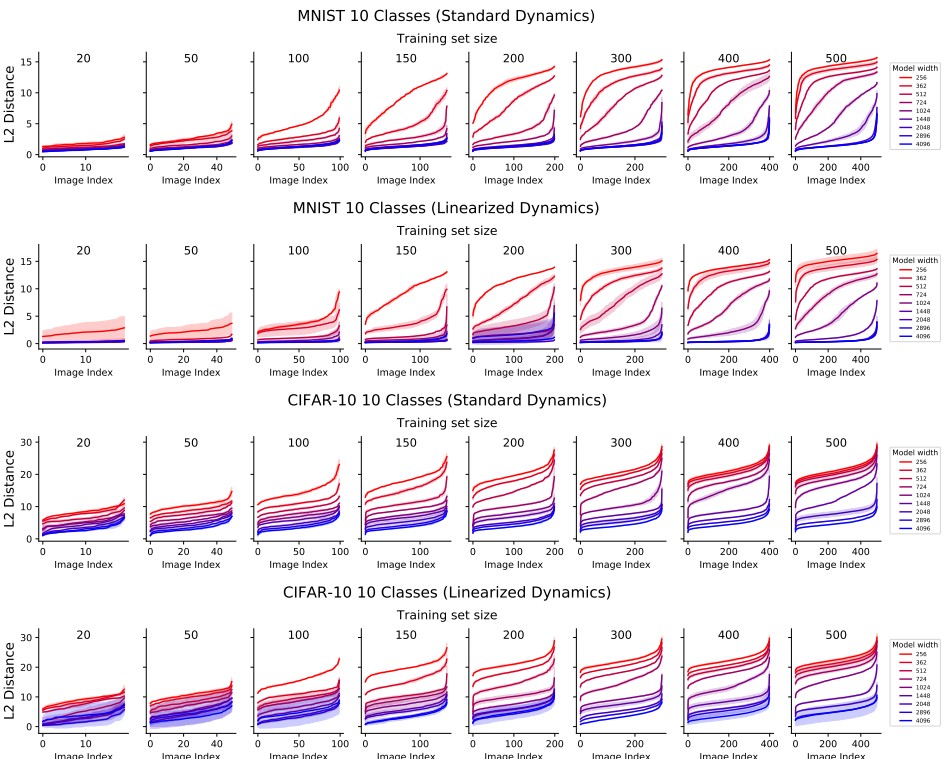

Figure 24: Reconstruction curves for multiclass classification

Figure 25: Reconstructions for MNIST Odd vs. Even, Linearized Dynamics, 4096 width.

Figure 26: Reconstructions for MNIST Odd vs. Even, Standard Dynamics, 4096 width.

Figure 27: Reconstructions for MNIST Odd vs. Even, Linearized Dynamics, 1024 width.

Figure 28: Reconstructions for MNIST Odd vs. Even, Standard Dynamics, 1024 width.

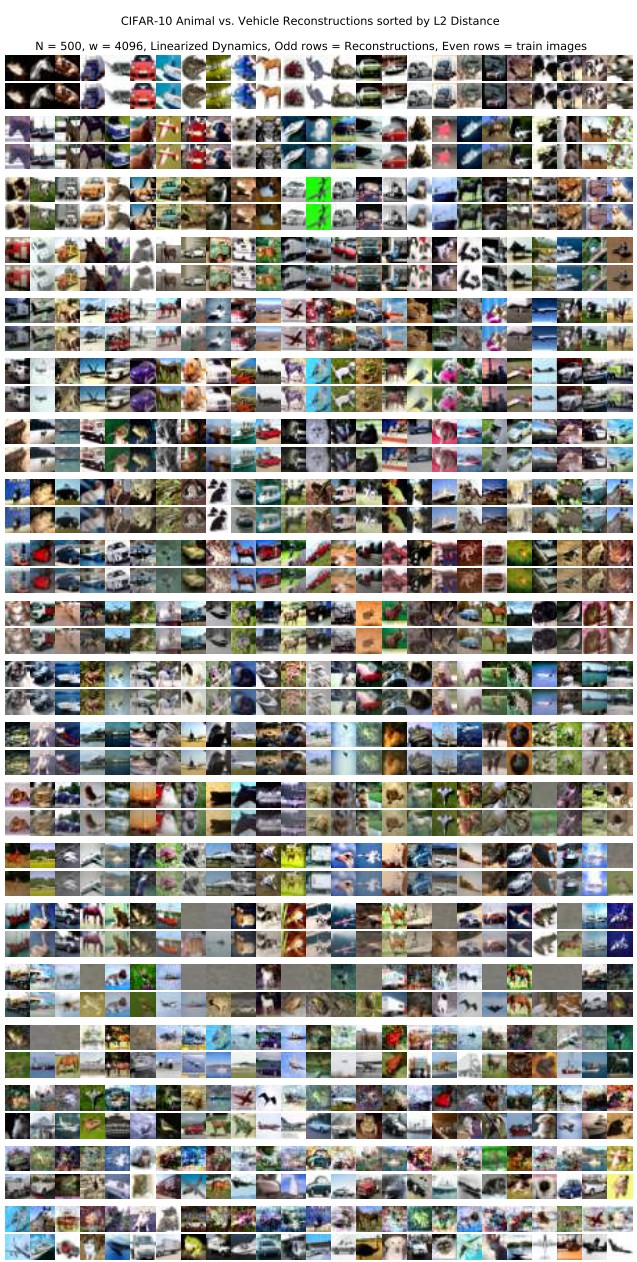

Figure 29: Reconstructions for CIFAR-10 Animal vs. Vehicle, Linearized Dynamics, 4096 width.

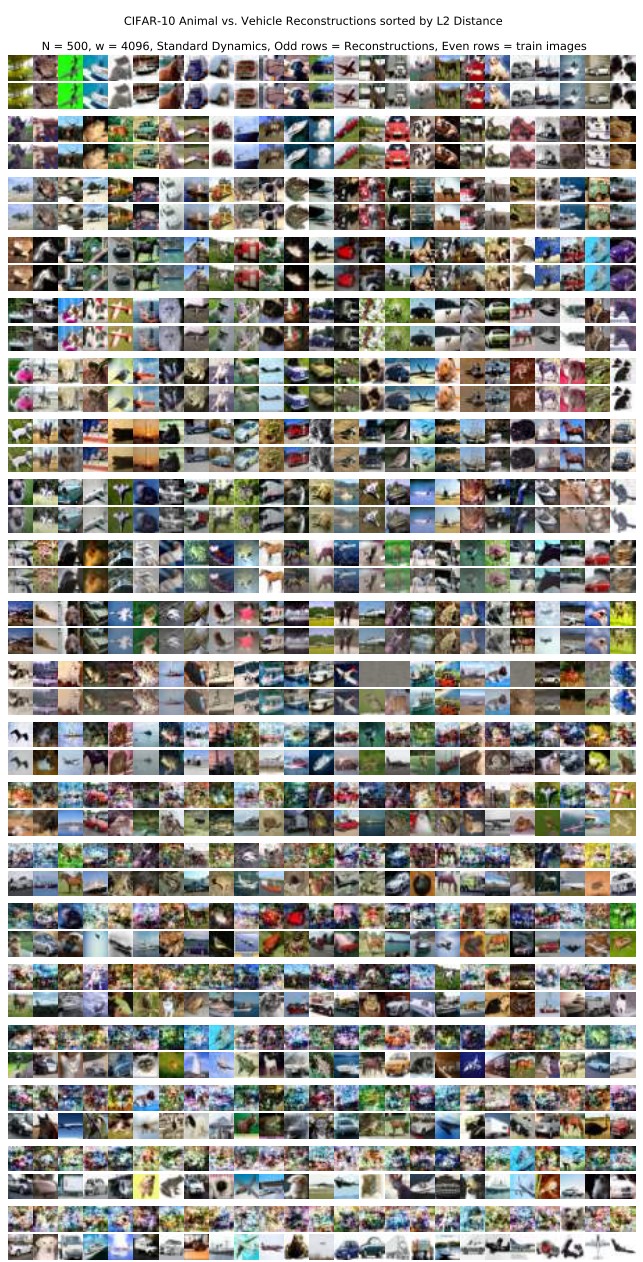

Figure 30: Reconstructions for CIFAR-10 Animal vs. Vehicle, Standard Dynamics, 4096 width.

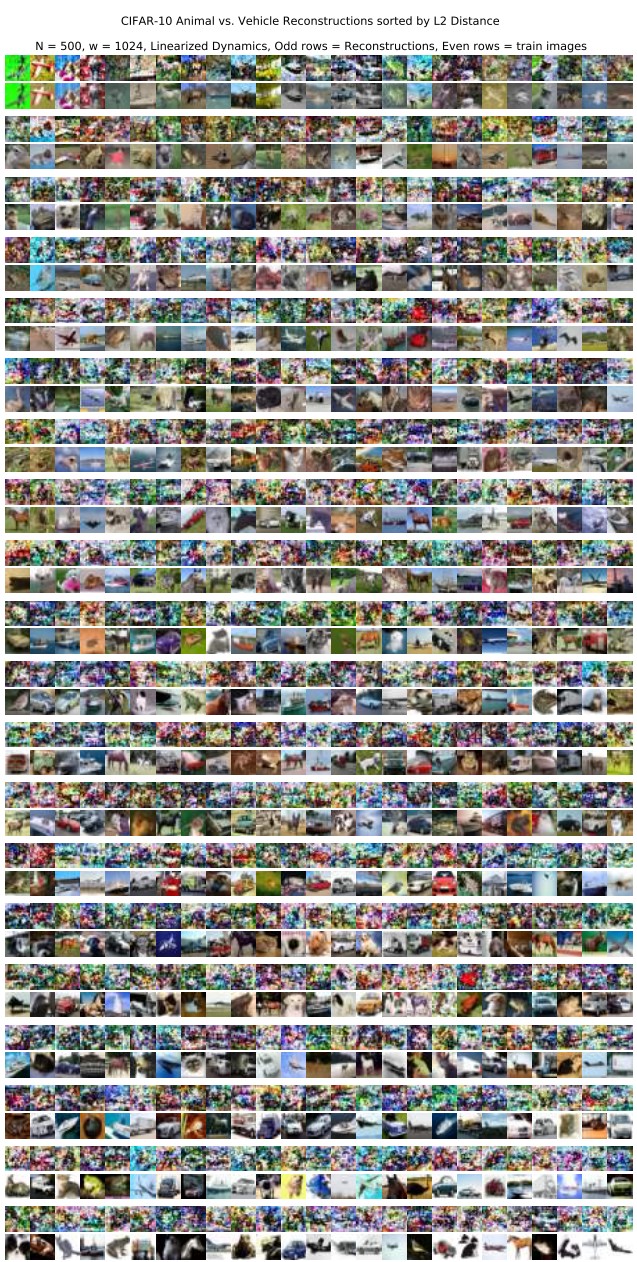

Figure 31: Reconstructions for CIFAR-10 Animal vs. Vehicle, Linearized Dynamics, 1024 width.

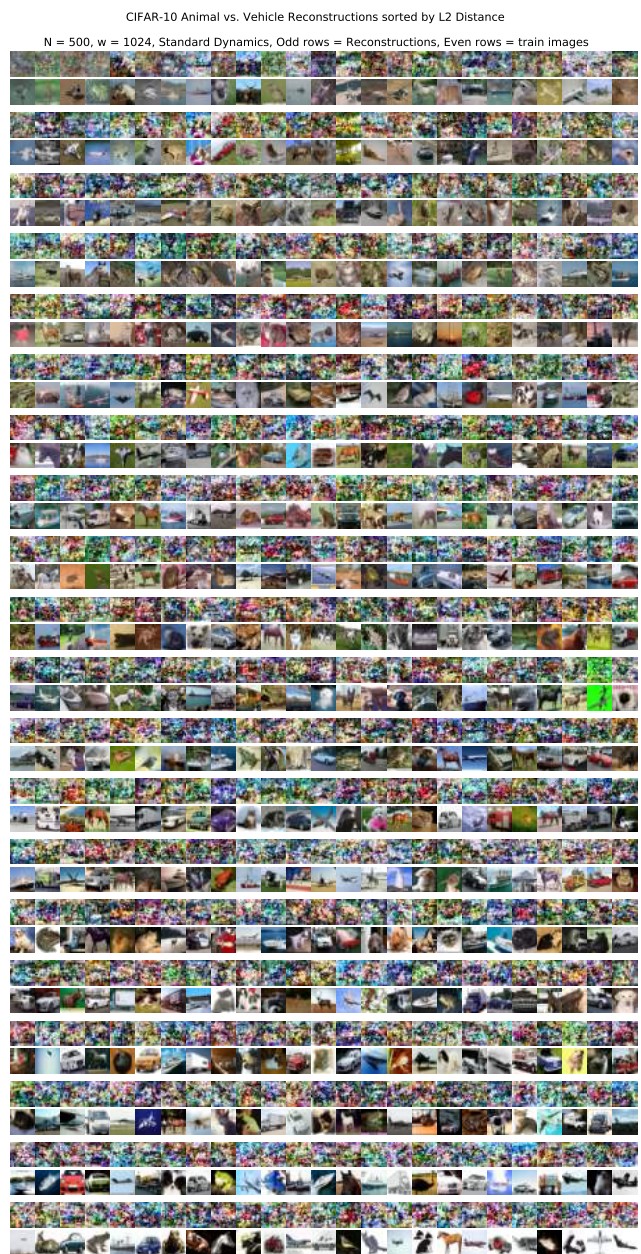

Figure 32: Reconstructions for CIFAR-10 Animal vs. Vehicle, Standard Dynamics, 1024 width.

Figure 33: Reconstructions for MNIST 10 Classes, Linearized Dynamics, 4096 width.

Figure 34: Reconstructions for MNIST 10 Classes, Standard Dynamics, 4096 width.

Figure 35: Reconstructions for MNIST 10 Classes, Linearized Dynamics, 1024 width.

Figure 36: Reconstructions for MNIST 10 Classes, Standard Dynamics, 1024 width.

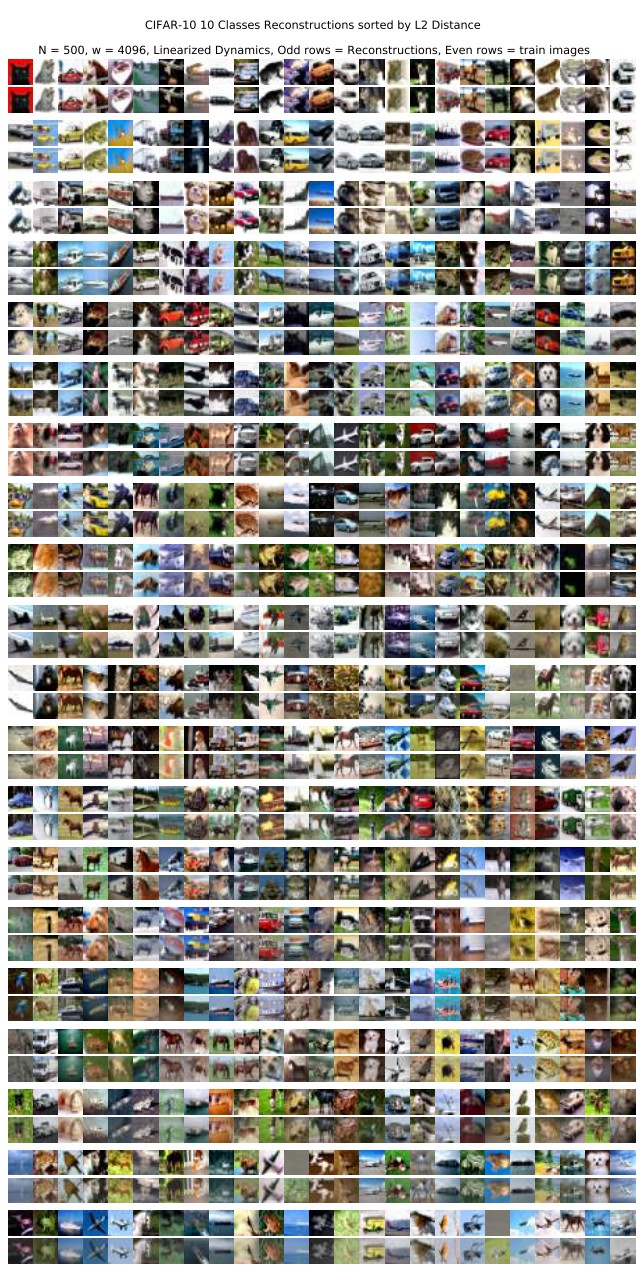

Figure 37: Reconstructions for CIFAR-10 10 Classes, Linearized Dynamics, 4096 width.

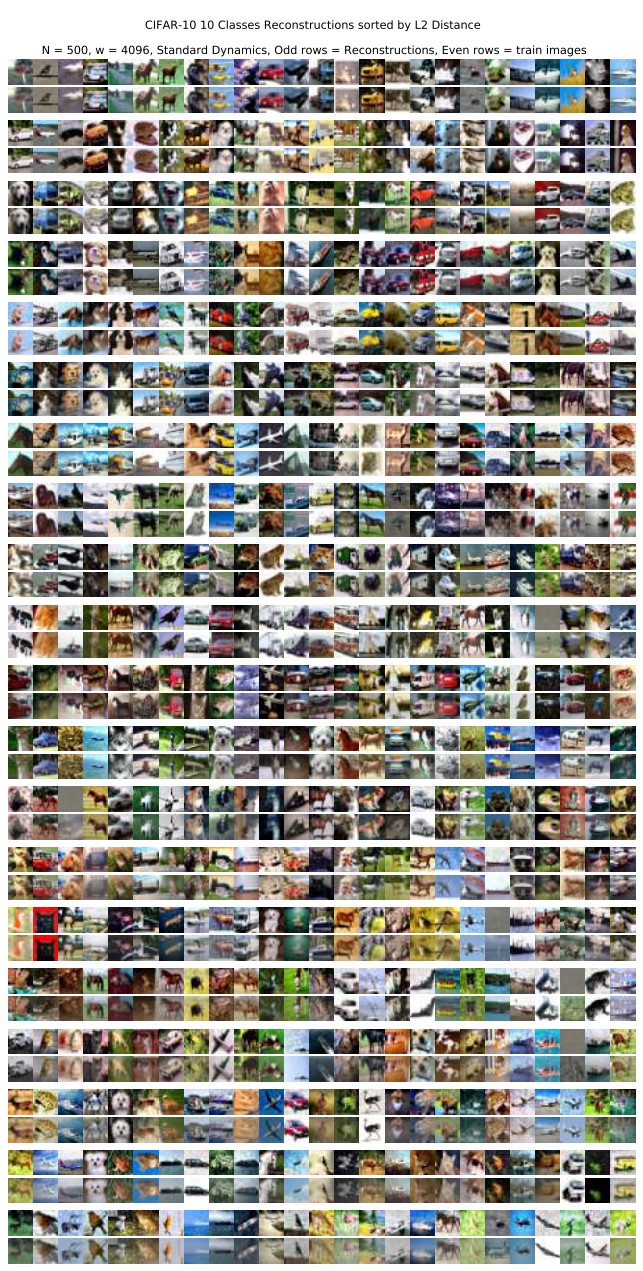

Figure 38: Reconstructions for CIFAR-10 10 Classes, Standard Dynamics, 4096 width.

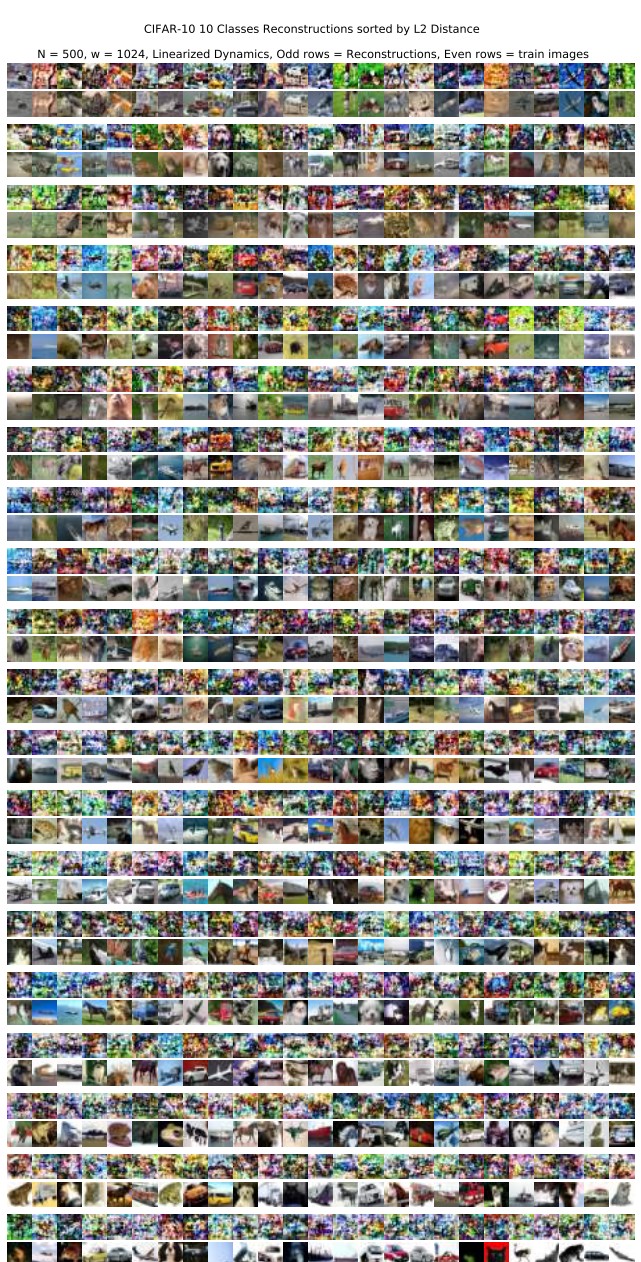

Figure 39: Reconstructions for CIFAR-10 A10 Classes, Linearized Dynamics, 1024 width.

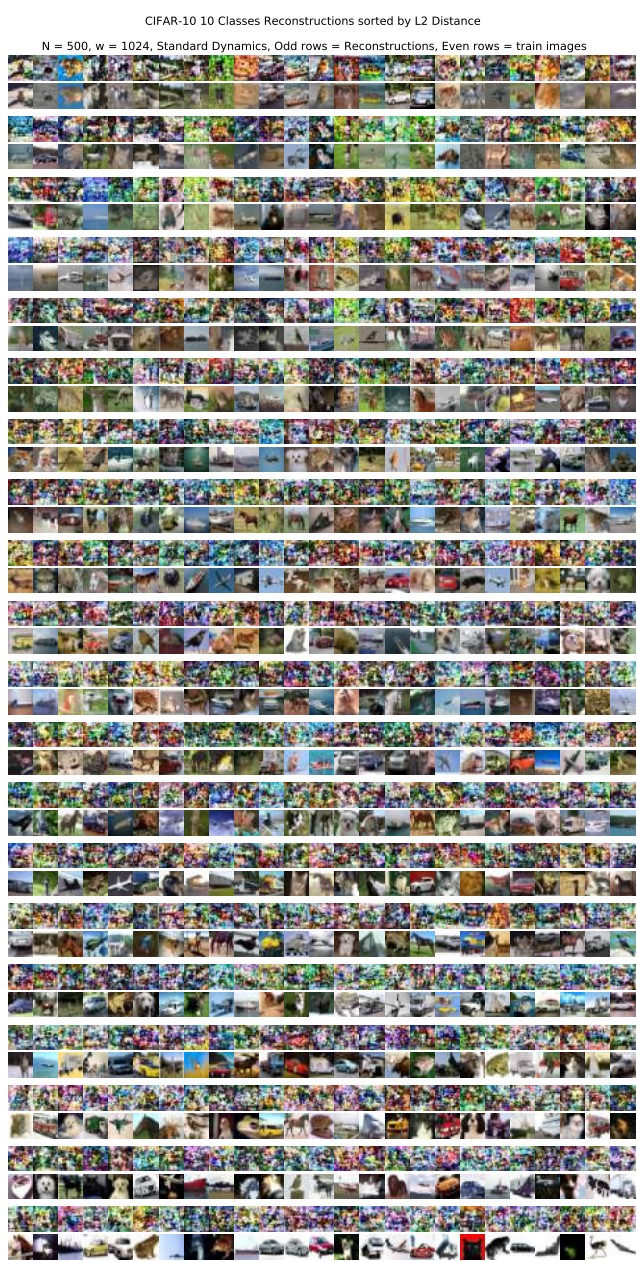

Figure 40: Reconstructions for CIFAR-10 10 Classes, Standard Dynamics, 1024 width.

