# OpenReview forum: "Understanding Reconstruction Attacks with the Neural Tangent Kernel and Dataset Distillation"
_ICLR.cc/2024/Conference — ICLR 2024 poster_

### Official Review · Reviewer_kua5 · 2023-10-29

**Soundness:** 2 fair
**Presentation:** 3 good
**Contribution:** 3 good
**Rating:** 6
**Confidence:** 3

**Summary:**

This paper studies the reconstruction attack, and shows that the reconstruction attack can recover all samples in the training data set. Studies are carried out on the properties of easily-reconstructed images.

**Strengths:**

This paper provides interesting experimental results. The main paper is clear and easy to understand.

**Weaknesses:**

My major concern is about the theoretical results in this paper. The paper claims their theoretical results as one of their major contributions, but from the presentation in the paper, this contribution is not as sound as the empirical side.

For the two theorems, Theorem 1 and 2, their presentation needs improvement.
*    For Theorem 1, one can intuitively understand its meaning, but it is hard to interpret the English sentence into a formal mathematical statement. The proof is also vague, with many descriptions but few math formulas and equations. It is hard to rigorously understand the mathematical meaning of this theorem, and it is also hard to check the correctness of the proof. The authors need to rewrite Theorem 1 (maybe leave an informal description in the main paper and postpone the full theory statement in the appendix) and provide a more rigorous proof.
*    For Theorem 2, the derivation in the appendix is readable, but the theorem statement in the main paper needs to be more clear. However, when considering infinite width of the neural network, some derivations are missing: for example, in Page 19, the first "->" needs more details. Although k_{\theta_0} -> k_{NTK}, the error terms are needed in the later derivation to show that a negligible |k_{\theta_0}-k_{NTK}| really leads to a negligible error term in the first "->" in Page 19.

**Questions:**

I think the empirical studies in this paper are sound but the theoretical parts are below the acceptance standard, so I give a score 5. Please consider improve the theorems, and update them either in the submission or reply in the rebuttal.

---

> ### Author Response · Authors · 2023-11-13
> **Comment Part 1**
>
> We thank the reviewer for their thoughtful critiques regarding the theory aspect of this paper, as well as recognizing the sound empirical findings. We would like to clarify the questions raised by the reviewer in this rebuttal, and these changes will make their way into the paper in future revisions.
>
> **Theorem 1**
>
> We will reword the theorem as follows:
>
> **Theorem 1**
>  If $L_\textrm{reconstruction} \to 0$ (from Eq. 7), as $w\to \infty $, then $X_R \to X_T $ in probability for training data, $X_T$ on the unit hypersphere.
>
> The proof follows closely with the outline provided in the text as is, but to clarify some bits which are possibly unclear, we rewrite the proof steps here:
> \begin{align}
>    L_{recon} =  &\quad \Big\|\Delta \theta - \sum_{\mathclap{\alpha_j x_j \in \alpha^R, X_R}} \alpha_j \nabla_{\theta_f} f_{\theta_f} (x_j)\Big\|^2_2 =\Big\|\sum_{\mathclap{\alpha_i, x_i \in \alpha^T, X_{T}}}{\alpha_i \nabla_{\theta_0}f(x_i)} - \sum_{\mathclap{\alpha_j x_j \in \alpha^R, X_R}} \alpha_j \nabla_{\theta_f} f_{\theta_f} (x_j)\Big\|^2_2 \\
>     &= \Big\|\sum_{\mathclap{\alpha_i, x_i \in \alpha^T, X_{T}}}{\alpha_i k_{\theta_0}(x_i, \cdot)} - \sum_{\mathclap{\alpha_j x_j \in \alpha^R, X_R}} \alpha_j k_{\theta_f}(x_j, \cdot)\Big\|^2_2
> &\to \Big\|\sum_{\mathclap{\alpha_i, x_i \in \alpha^T, X_{T}}}{\alpha_i k_{NTK}(x_i, \cdot)} - \sum_{\mathclap{\alpha_j x_j \in \alpha^R, X_R}} \alpha_j k_{NTK}(x_j, \cdot)\Big\|^2_2 \mathbf{i.p.}
> \end{align}
> Define
> \begin{align}
> P_T = \sum_{\alpha_i, x_i \in \alpha^T, X_{T}}{\alpha_i\delta(x_i)},
> P_R = \sum_{\alpha_j, x_j \in \alpha^R, X_{R}}{\alpha_j\delta(x_j)}
> \end{align}
>
> Which are both signed measures over $\Omega = S^{d}$.
>
> Then we can rewrite
> \begin{align}
> \sum_{\mathclap{\alpha_i, x_i \in \alpha^T, X_{T}}}{\alpha_i k_{NTK}(x_i, \cdot)}
> \end{align} as
> \begin{align}
> \int_{\Omega} k_{NTK}(x, \cdot) dP_T(x)
> \end{align}, and likewise for $P_R$. Then, we have
>
> \begin{align}
> L_{recon} \to \Big\|\int_{\Omega} k_{NTK}(x, \cdot) dP_T(x) - \int_{\Omega} k_{NTK}(x, \cdot) dP_R(x)\Big\|^2_2 = MMD_{k_{NTK}}(P_R, P_T)
> \end{align}
>
> This is the maximum mean discrepancy described in [1]. From [2], we have that $k_{NTK}$ being universal over $S^{d}$ [3] implies that i $MMD_{k_{NTK}}(P_R, P_T) = 0$ implies that $P_R = P_T$, i.e. $X_R = X_T$. $L_{recon} \to 0$, as assumed, so $MMD_{k_{NTK}}(P_R, P_T) \to 0$, in probability. As $k_{NTK}$, and the MMD are continuous in their inputs, this implies that $X_R \to X_T$ in probability as well.
>
> We hope that this clarifies the proof. We acknowledge that the description in the text is rather space constrained and in future revisions we will have the more informal statement in the main text, and defer the proof to the appendix, as suggested by the reviewer.
>
> [1]Arthur Gretton, Karsten M. Borgwardt, Malte J. Rasch, Bernhard Sch ̈olkopf, and Alexander Smola.
> A kernel two-sample test.
>
> [2] Jacot, A., Gabriel, F., & Hongler, C. (2018). Neural Tangent Kernel: Convergence and Generalization in Neural Networks.
>
> [3] Bharath K. Sriperumbudur, Kenji Fukumizu, and Gert R.G. Lanckriet. Universality, characteristic
> kernels and rkhs embedding of measures.
>
> [4] Timothy Nguyen, Zhourong Chen, and Jaehoon Lee. Dataset meta-learning from kernel ridge-
> regression.

---

> > ### Author Response · Authors · 2023-11-13
> > **Comment Part 2**
> >
> > **Theorem 2**
> >
> > The theorem as stated in the text currently is not quite correct, and should read:
> >
> > **Theorem 2** The **infinite-width variant** of reconstruction scheme of Eq. 4 with KKT points of Eq. 5 and Eq. 6, with M ≤ N where M is the reconstruction image counts and N is the dataset size, can be written as a kernel inducing point distillation loss under a different norm **plus a non-negative correction term $\lambda$** as follows:
> >
> > \begin{align}
> > L_{recon} &= \|y_T - K_{TR}K_{RR}^{-1}y_R\|^2_{K^{-1}_{TT}}  + \lambda
> > \end{align}
> >
> > Note that in this, we assume that we are already in the infinite width regime, i.e. we don’t need to take limits in the proof for this because we already assume we are working with the infinite width kernels. The details of taking in limit are in [2], and are used in the proof of Thm. 1. Note that this assumption makes sense as the KIP algorithm already works with the infinite width kernels [4], so it would not make much sense to compare a finite-width version of ours to the infinite version of theirs. Also note that $\lambda$ is **not** strictly an error term, but a non-negative correction term which we describe in the appendix. So rather it should be taken that the RKIP loss *upper bounds* a KIP-like loss, rather than equalling in. Also we note an error on page 18 equation 2 where we omitted the $\lambda$ term in the RKIP loss.
> >
> > Again we would like to thank the reader for their careful reading of the paper and attention to the rigor of our theory. We would be happy to discuss more details, and will update the manuscript once the reviewer is satisfied with the clarifications.
> >
> > [1]Arthur Gretton, Karsten M. Borgwardt, Malte J. Rasch, Bernhard Sch ̈olkopf, and Alexander Smola.
> > A kernel two-sample test.
> >
> > [2] Jacot, A., Gabriel, F., & Hongler, C. (2018). Neural Tangent Kernel: Convergence and Generalization in Neural Networks.
> >
> > [3] Bharath K. Sriperumbudur, Kenji Fukumizu, and Gert R.G. Lanckriet. Universality, characteristic
> > kernels and rkhs embedding of measures.
> >
> > [4] Timothy Nguyen, Zhourong Chen, and Jaehoon Lee. Dataset meta-learning from kernel ridge-
> > regression.

---

> > > ### Comment · Reviewer_kua5 · 2023-11-14
> > > **Followup in the proof of Theorem 1**
> > >
> > > I appreciate the authors clarify the theory part. For Theorem 1, could you elaborate more on why both $f_{\theta_0}$ and $f_{\theta_f}$ converges to the same kernel? Do you use existing results in literature?

---

> > > > ### Author Response · Authors · 2023-11-14
> > > >
> > > > Yes, these are results in previous literature. Specifically, $k_{\theta_0}$ converging to $k_{NTK}$ is given by Theorem 1 in [1], and corollary 2.4 in [3]. Likewise $k_{\theta}$ staying constant over the course of training (so that $k_{\theta_f}$ also converges to $k_{NTK}$) is given in Theorem 2 in [1] and theorem 2.1 in [2]. We will cite these theorems more specifically in future revisions.
> > > >
> > > > [1] Jacot, A., Gabriel, F., & Hongler, C. (2018). Neural Tangent Kernel: Convergence and Generalization in Neural Networks.
> > > >
> > > > [2] Lee, J., Xiao, L., Schoenholz, S. S., Bahri, Y., Novak, R., Sohl-Dickstein, J., & Pennington, J. (2020). Wide neural networks of any depth evolve as linear models under gradient descent
> > > >
> > > >
> > > > [3] Greg Yang. Scaling limits of wide neural networks with weight sharing: Gaussian process behavior, gradient independence, and neural tangent kernel derivation.

---

> > > > > ### Comment · Reviewer_kua5 · 2023-11-14
> > > > >
> > > > > Thanks, I have raised my score from 5 to 6. Please help update all the changes in the paper, and also add the references when using the convergence of NTK.

---

> > > > > > ### Author Response · Authors · 2023-11-18
> > > > > >
> > > > > > We would like to thank the reviewer for considering our rebuttal and raising their score. We will incorporate the revisions in to the main text once we have collected feedback from other reviewers.

---

### Official Review · Reviewer_GBp6 · 2023-10-30

**Soundness:** 4 excellent
**Presentation:** 3 good
**Contribution:** 3 good
**Rating:** 8
**Confidence:** 4

**Summary:**

In this paper, the authors investigated the reconstruction attack in the view of neural tangent kernel (NTK). From a well formulated description, the authors showed interesting results with both theoretical and practical meaning.

**Strengths:**

The memory of deep neural networks is of great interests and importance. It is believed that the memory and also the memorization is closely related to the training dynamics, which is however not clearly investigated. Thus, I personally like the way of modelling the data reconstruction by NTK, which indeed capture the main properties on memorization dynamics. Though there are still many simplification, the good performance demonstrate the rationality of the modelling. So I think the main strengths include:

- A novel and interesting way of modelling memorization from training dynamics.

- Theoretical discussion well coincides with numerical experiments.

- Clear discussion on the weakness, which actually could inspire future works.

**Weaknesses:**

The main weakness are for some unclear settings. Please see the questions below.

**Questions:**

In the current version, the reconstruction performance is related to the number of training samples as well as the property of the samples. How about the effect of data dimension. Especially, the author cast the reconstruction loss as a sparse coding, also Haim et al. (2021) regarded the training process as encoding. Then, can the authors obtain some conclusion about data dimension?

The reconstruction problem is a complicated optimization problem and the result could be totally different when different initial solutions are used. I notice that in algorithm 2 "randomly initialized reconstruction images" are used. Then how about the divergence of the reconstruction result? Is that necessary to use special initialization, e.g., an image in one of the two classes, an image from another class, a natural image of which the class is not in the training set, or a random generated matrix?

After reading the rebuttal and good discussion, I would like to increase the score from 6 to 8. But please talk more about the link to e.g., GradViT: Gradient Inversion of Vision Transformers; Deep Leakage from Gradients, which can use local gradient information (even the model is not well trained) to reconstruct the training samples.

---

> ### Author Response · Authors · 2023-11-13
>
> We would like to thank the reviewer for acknowledging the contributions of this paper in linking the training dynamics of neural networks with the memorization phenomenon, and in recommending its acceptance. We would like to address the questions raised by the reviewer here:
>
> **Data Dimension** Indeed the efficacy of our reconstruction attack depends on the dimension of the data. Previous work has shown that the quality of the NTK approximation suffers with high data dimension [1,2], and as our method relies on the NTK approximation to prove theorem 1, we expect our attack to be less effective with larger data dimension. Indeed, in appendix J.2. (page 26) and figure 18 and 19 (page 25), we perform the same attack on a higher resolution dataset Tiny-Imagenet (resolution 64x64x3). We see that reconstruction quality suffers, but we are still able to reconstruct images with high quality. Combining the approach provided in this paper with prior work on understanding quality of NTK approximation [1,2] could lead to a more robust definition of how networks “encode” their data points, which is the subject of future work.
>
>
> **Image Initialization** In all our experiments we initialize reconstruction images with random noise, ensuring that we have no a priori knowledge about the dataset. We will clarify this in future revision. We observe that our method is not sensitive to the initialization scheme in practice, although we acknowledge that the reconstruction loss is provably non-convex, but in practice this does not seem to be a major concern.
>
> [1] Bombari, S., Amani, M. H., & Mondelli, M. (2023). Memorization and Optimization in Deep Neural Networks with Minimum Over-parameterization.
>
> [2] Adlam, B., & Pennington, J. (2020). The Neural Tangent Kernel in High Dimensions: Triple Descent and a Multi-Scale Theory of Generalization.

---

> ### Author Response · Authors · 2023-11-18
>
> We were wondering if the reviewer has had a chance to consider our rebuttal, and if there were still any remaining concerns which we can address before the discussion period ends.

---

> > ### Comment · Reviewer_GBp6 · 2023-11-19
> > **thanks and one more question**
> >
> > Thanks for the authors' discussion on the dimensionality and initialization. Both I think there are lots of interesting things to think. For example, the learning behaviour of DNN is closely linked to different initializations.
> >
> > One more question is about the link to approaches that can reconstruct images from gradients: e.g., GradViT: Gradient Inversion of Vision Transformers; Deep Leakage from Gradients. Is that the case that they use local gradient but you use the "global one", i.e., the gradient of the linearly approximated system.

---

> > > ### Author Response · Authors · 2023-11-19
> > >
> > > Yes, in the limit of a single gradient step our attack would correspond to gradient inversion attacks [1,2], and is an interesting point raised by the reviewer. With this connection in mind, Theorem 1 does actually guarantee the exactness of gradient inversion attacks in the infinite width regime. This is because our proof does not require training to convergence, and only relies on the universal nature of the NTK [3], which still holds even in early stopping. For more details on the proof, see also the comment chain with reviewer kua5. Additionally, we have experiments investigating the effect of early stopping in appendix G.1, so we can roughly expect the quality of gradient inversion attacks to correspond to the extreme early-stopping limit. We can elaborate more on this connection in the next revision of the text. Note however that our proof does not necessarily make any statements of the efficacy of gradient inversion for non-infinite width networks, as the learned finite-width neural tangent kernel may not have the same universal property of the initialization infinite-width NTK.
> > >
> > >
> > > [1] Ligeng Zhu, Zhijian Liu, , and Song Han. Deep leakage from gradients. In Annual Conference on Neural Information Processing Systems (NeurIPS), 2019.
> > >
> > > [2] Hatamizadeh, A., Yin, H., Roth, H., Li, W., Kautz, J., Xu, D., & Molchanov, P. (2022). GradViT: Gradient Inversion of Vision Transformers.
> > >
> > > [3] Jacot, A., Gabriel, F., & Hongler, C. (2018). Neural Tangent Kernel: Convergence and Generalization in Neural Networks.

---

> > > > ### Comment · Reviewer_GBp6 · 2023-11-21
> > > > **thanks**
> > > >
> > > > Thanks for your further explanation. Gradient inversion attacks can reconstruct training samples even when the NN has not converged (the quality is lower than using those attack on well-trained NN). At least, it empirically shows the reconstruction may come from local properties rather than the whole training process. I just write this idea here since may be there is a bit difference to the discussions in this paper. BUT I do not to say this paper is not good. On contrast, I suggest acceptance of this paper. So I would like to increase the score to 8.

---

### Official Review · Reviewer_FNQS · 2023-11-08

**Soundness:** 3 good
**Presentation:** 2 fair
**Contribution:** 3 good
**Rating:** 6
**Confidence:** 2

**Summary:**

This paper introduces a new, more stable method for performing reconstruction attacks against neural networks based on the neural tangent kernel. This method requires access to both the initial weight state and the trained weight state, but does not require any information about the data distribution. A number of ablation studies confirm conventional wisdom that larger networks essentially memorize their training sets, that outlier datapoints are most vulnerable to reconstruction attacks, and that there is a strong mathematical connection between reconstruction attacks and dataset distillation.

**Strengths:**

The paper develops a very nice algorithm for dataset distillation based on inducing point methods. Figure 8 in particular shows the value of the proposed approach. The underlying theoretical connections to dataset inversion provide confidence in the method.

The paper provides thorough experimental evidence for the theoretical claims. The datasets used (restricted MNIST and CIFAR) are small, but this is to be expected with a computationally-intensive object like the NTK. The reconstruction attacks on ImageNet are impressive.

The discussion about the choice of kernel in Appendix I is interesting. The authors should at least incorporate the insight about combining initial and final weight states into the main text. Likewise, why are the results on more complex architectures reserved for Appendix J? These are interesting results that should be incorporated into the paper, if even briefly.

**Weaknesses:**

The authors claim that “outliers” are more vulnerable to reconstruction attacks, but this notion of “outlier” is not well defined up front. I believe that your technical definition for outliers is points that have high $\alpha$ values (i.e. points that are “hard to fit”), but this does not necessarily mean that these are points that are distant e.g. under the Euclidean metric in the input space.

The text on most of the figures is so small it makes them hard to read, even on a screen.

Reconstruction plots are not explained until the first paragraph of Page 6, but the plots appear as early as page 4. This creates a readability problem, since the construction of these plots is not self-evident.

It would be nice to have a central definition of all model variants (e.g. RKIP, RKIP-finite). These definitions are currently spread across the paper.

Small issues:
* This phrase in the abstract doesn’t make sense: “of its effective regime **which** datapoints are susceptible to reconstruction.”
* The statement that non-privacy preserving networks are “useless” is probably a bit hyperbolic in the introduction, this is very application-dependent.
* In Table 1, specify whether the values are “accuracy”.

**Questions:**

1. Can you make claims about the effect of self-supervised learning on the effectiveness of your reconstruction attacks? Does your method naturally extend to structured outputs, not just scalar outputs?
2. Do you have any speculation how informative your results will be for networks trained with different losses than MSE? Is this a simplifying assumption to enable your proofs or a factor that could significantly change the structure of the empirical NTKs your method uses?
3. Is there an underlying assumption in (1) that there is a prior (i.e. weight decay) on the parameters of the network?

---

> ### Author Response · Authors · 2023-11-13
>
> We would like to thank the reviewer for acknowledging our contributions and recommending our acceptance. We would like to take this space to address some of their concerns.
>
> **Various presentation points**
> We apologize for the poor readability of some of the figures. Indeed we were very constrained in terms of space, and consequently had to reduce figure sizes, as well as move certain results to the appendix, which is why the results on convolutional architectures were moved there. We will address these concerns, along with the other minor concerns pointed out by the reviewer in future revisions.
>
> **Definition of outliers**
> We define an outlier to be a datapoint which is harmful/uninformative to the training procedure, has a strange/unusual visual appearance or is far away from other training data in terms of training dynamics. Moreover, there is no agreed-upon definition in the literature. We show that easily reconstructed images fit these three properties as follows:
> 1. Harmful to the training procedure/uninformative - see fig. 6, in which we show that removing these data points do not adversely affect training
> 2. Strange visual appearance - see figures 25-40 in the appendix, where we see that the more easily reconstructed images tend to have more “unusual” features (i.e. solid-colored backgrounds for CIFAR-10, faint lines for MNIST)
> 3. Difficulty to fit - see the discuss of alpha values
> We do not believe that L2 distance in the euclidean space is a good definition for an “outlier” data point when working with neural networks, as it does not take into account data structure, or the training procedure. We believe that the qualities that we looked at in terms of defining an outlier are suitable properties and reflect a more nuanced view which takes into account training dynamics and dataset properties. We are happy to discuss this further.
>
> **Self Supervised Learning and output structure**
> We note that our attack works with vector outputs, as seen by the results on multiclass classification. Extended our work to self-supervised learning may be more difficult as self-supervised learning typically occurs outside of the NTK regime, i.e. there is a large amount of representation learning, which is not modeled by the NTK. However, we believe that our work fills a vital first-step in understanding reconstruction attacks in a more simplified regime.
>
> **Non-MSE losses**
> Our method should work on non-MSE losses such as cross-entropy. We include a discussion in appendix G.2. The main requirement of our attack to work is that the change in network parameters lies in the span of the datapoint gradients, and that these gradients do not change significantly over the course of training. This is fulfilled by wide neural networks under any loss (see [1], appendix B for more details). We chose to focus on MSE loss in this work as it allows a closed for solution for the $\alpha$ values, simplifying analysis, but the analysis still applies. In the proof for Theorem 1, we do not explicitly require MSE loss, so this proof still holds for non-MSE losses in the kernel regime.
>
> **Weight Decay**
> We do not assume weight decay, but including weight decay would not affect the theoretical results in our paper. Like in the answer to the previous question, weight decay does not affect whether the weight changes remain in the span of the dataset gradients, and thus our analysis still holds. Note that we can have a similar effect to weight decay with early stopping, and we carefully ablate the effects of early stopping in appendix G.1.
>
> We hope that these points address the reviewers main concerns and we would be happy to discuss further with them.
>
> [1] Lee, J., Xiao, L., Schoenholz, S. S., Bahri, Y., Novak, R., Sohl-Dickstein, J., & Pennington, J. (2020). Wide neural networks of any depth evolve as linear models under gradient descent

---

> > ### Comment · Reviewer_FNQS · 2023-11-20
> > **Thank you for your reply**
> >
> > I appreciate the authors' detailed response. This is an interesting paper. I would like to keep my score of 6.

---

### Meta-Review · Area_Chair_iYmd · 2023-12-04

**Metareview:**

The authors described the connection between the loss in kernel ridge regression (with the formulation with kernel inducing points) for data distillation and the loss for reconstruction attacks (the loss similiar to Haim et al 2022). When the kernels are infinitely wide neural tangent kernels, the reconstruction loss becomes MMD between the distributions of the training and reconstruction data sets, respectively. Due to the universality of the NTK (but is it true for any NTKs? not sure), the MMD becomes a metric and as a result it is possible to reconstruct the entire training data. Relying on NTKs being universal kernels, theorem 1 seems straightforward, while probably this is the first time this connection was noticed/revealed. What seems more impactful is how their analysis applies to finite-width NTK regimes. The authors empirically studied this problem in various settings, which seems a real strength of this paper. All three reviewers agreed that this paper makes a concrete contribution and suggested acceptance.

It would be also nice to add something related: how differential privacy (if noise is added for differential privacy during training) changes the outcome of reconstruction attacks in the same NTK settings (both finite-width and infinite-width NTK regimes).

**Justification For Why Not Higher Score:**

The connection between the KRR loss in KIP and the loss for reconstruction attacks does not look too surprising to my eyes. Perhaps poster presentation is more appropriate based on their findings in the paper.

**Justification For Why Not Lower Score:**

All three reviewers are giving higher than 6. There are no obvious reasons to object to them.

---

### Decision · Program_Chairs · 2024-01-16

Accept (poster)